# Statin-mediated reduction in mitochondrial cholesterol primes an anti-inflammatory response in macrophages by upregulating Jmjd3

Zeina Salloum[1], Kristin Dauner[1], Yun-feng Li[2], Neha Verma[1], David Valdivieso-González[3,4], Víctor Almendro-Vedia[3,4], John D Zhang[1], Kiran Nakka[5], Mei Xi Chen[5,6], Jeffrey McDonald[7], Chase D Corley[7], Alexander Sorisky[1,8], Bao-Liang Song[2], Iván López-Montero[3,4], Jie Luo[2], Jeffrey F Dilworth[5,6,8], Xiaohui Zha[1,9]*

[1]Chronic Disease Program, Ottawa Hospital Research Institute, Ottawa, Canada; [2]College of Life Sciences, Wuhan University, Wuhan, China; [3]Departamento Química Física, Universidad Complutense de Madrid, Avda, Madrid, Spain; [4]Instituto de Investigación Biomédica Hospital Doce de Octubre (imas12), Madrid, Spain; [5]Sprott Center for Stem Cell Research, Regenerative Medicine Program, Ottawa Hospital Research Institute, Ottawa, Canada; [6]Department of Cell and Regenerative Biology, University of Wisconsin, Madison, United States; [7]Department of Molecular Genetics, The University of Texas Southwestern Medical Center, Dallas, United States; [8]Department of Cellular and Molecular Medicine, University of Ottawa, Ottawa, Canada; [9]Departments of Medicine and of Biochemistry, Microbiology & Immunology, University of Ottawa, Ottawa, Canada

*For correspondence:
xzha@ohri.ca

Competing interest: The authors declare that no competing interests exist.

**Abstract** Statins are known to be anti-inflammatory, but the mechanism remains poorly understood. Here, we show that macrophages, either treated with statin in vitro or from statin-treated mice, have reduced cholesterol levels and higher expression of *Jmjd3*, a H3K27me3 demethylase. We provide evidence that lowering cholesterol levels in macrophages suppresses the adenosine triphosphate (ATP) synthase in the inner mitochondrial membrane and changes the proton gradient in the mitochondria. This activates nuclear factor kappa-B (NF-κB) and *Jmjd3* expression, which removes the repressive marker H3K27me3. Accordingly, the epigenome is altered by the cholesterol reduction. When subsequently challenged by the inflammatory stimulus lipopolysaccharide (M1), macrophages, either treated with statins in vitro or isolated from statin-fed mice, express lower levels proinflammatory cytokines than controls, while augmenting anti-inflammatory *Il10* expression. On the other hand, when macrophages are alternatively activated by IL-4 (M2), statins promote the expression of *Arg1*, *Ym1*, and *Mrc1*. The enhanced expression is correlated with the statin-induced removal of H3K27me3 from these genes prior to activation. In addition, *Jmjd3* and its demethylase activity are necessary for cholesterol to modulate both M1 and M2 activation. We conclude that upregulation of *Jmjd3* is a key event for the anti-inflammatory function of statins on macrophages.

## Editor's evaluation

The manuscript by Salloum and colleagues examines the role of statin-mediated regulation of mitochondrial cholesterol as a determinant of epigenetic programming via JMJD3 in macrophages. The findings could be valuable for understanding statin-mediated anti-inflammatory functions in

macrophages. The evidence for a statin-cholesterol-JMJD3-H3K27 axis as a regulator of macrophage function is solid.

## Introduction

Many chronic diseases, including atherosclerosis, are associated with a low-grade level of inflammation (*Tabas and Glass, 2013*). In atherosclerosis, macrophages are overloaded with cholesterol and are inflamed, which causes lesion formation (*Tabas and Lichtman, 2017*). Statins, by inhibiting cholesterol biosynthesis and upregulating low-density lipoprotein (LDL) receptor expression in hepatocytes and other cells, lower the level of circulating LDL. This leads to a decrease in the amount of cholesterol in peripheral tissues including macrophages (*Levy et al., 1992*; *Hochgraf et al., 1994*; *Lijnen et al., 1994*; *Parihar et al., 2014*). Statins reduce inflammation, but the mechanism for this effect is yet to be defined (*Libby, 2021*). Previously, we and others have observed that the level of cholesterol in resting macrophages, i.e., prior to stimulation, correlates directly with their pattern of inflammatory activation (*Ma et al., 2012*; *Yvan-Charvet et al., 2008*; *Zhu et al., 2008*). For instance, when encountering lipopolysaccharides (LPS), cholesterol-rich macrophages release more inflammatory cytokines, such as TNF-α, IL-6, and IL12p40, and less anti-inflammatory cytokine IL-10, relative to control macrophages. Conversely, macrophages with reduced cholesterol content, such as those expressing ABCA1/G1 or being treated with statins, express fewer inflammatory cytokines, and more IL-10 upon identical LPS exposure (*Ma et al., 2012*; *Yvan-Charvet et al., 2008*; *Zhu et al., 2008*). Noticeably, this association of cholesterol with specific type of macrophage inflammation is observed with multiple inflammatory stimuli, including ligands to TLR2, 3, 4, and other TLRs (*Yvan-Charvet et al., 2008*), implying a shared background in resting macrophages. It is known that, in order to mount a timely and vigorous defence against pathogens, macrophages activate several hundred genes immediately after sensing danger signals (*Raetz et al., 2006*). This is achieved by employing a few select signal-dependent transcription factors, such as NF-κB, on a genome that is largely poised, i.e., epigenetically configured, prior the stimulation (*Link et al., 2015*). The organization of the epigenome in macrophages is initially defined by lineage-determining transcription factors and subsequently by metabolic/environmental cues, including those experienced in the past (*Link et al., 2015*; *Troutman et al., 2021*). This forms a poised state in resting macrophages and allows the expression of context-dependent inflammatory genes (*Spann et al., 2012*; *Ivashkiv, 2013*). We therefore speculate that cholesterol levels may directly influence the epigenome in resting macrophages, thereby influencing the inflammatory phenotype upon activation.

## Results

### Reducing cholesterol levels in macrophages activates the NF-κB pathway and upregulates Jmjd3, a histone demethylase

To test whether cholesterol levels alone can produce factors that change the epigenetic configuration in macrophages, we first treated RAW macrophages with statins. Lovastatin, compactin, simvastatin, and pravastatin were used exchangeably throughout the study without significant differences in results. Cells were treated with statins for 2 days, reducing cellular cholesterol by about 30% (*Ma et al., 2012*; *Mayor et al., 1998*). We also used methyl-β-cyclodextrin (MCD) 1-hr treatment to acutely reduce cholesterol by a similar amount (*Ma et al., 2012*). This was to verify the cholesterol specificity of statins' effects and also to provide a 1-hr protocol for inhibitor studies. RNA-seq was performed at the end of statin- or MCD-based treatments. As shown in *Figure 1A*, macrophages treated with statins altered the expression of a large number of genes (log$_2$ >1) (*Figure 1A, a*; *Supplementary file 1*; the top 40 upregulated genes: *Figure 1—figure supplement 1A*). We applied the gene set enrichment analysis (GSEA) to all differentially expressed genes (DEGs; up- and downregulation), using the Hallmark genes database (*Figure 1A, b*). Genes of the NF-κB pathway activated by TNFα were the most highly represented group upon statin treatment (*Figure 1A, b, c*). Similarly, MCD-treated macrophages showed an identical, highly represented group, i.e., NF-κB pathway activated by TNFα (*Figure 1B, a–c*; *Supplementary file 2*). The top 40 upregulated genes are in *Table 1*. This analysis suggests that cholesterol depletion induces activation of NF-κB pathways in macrophages.

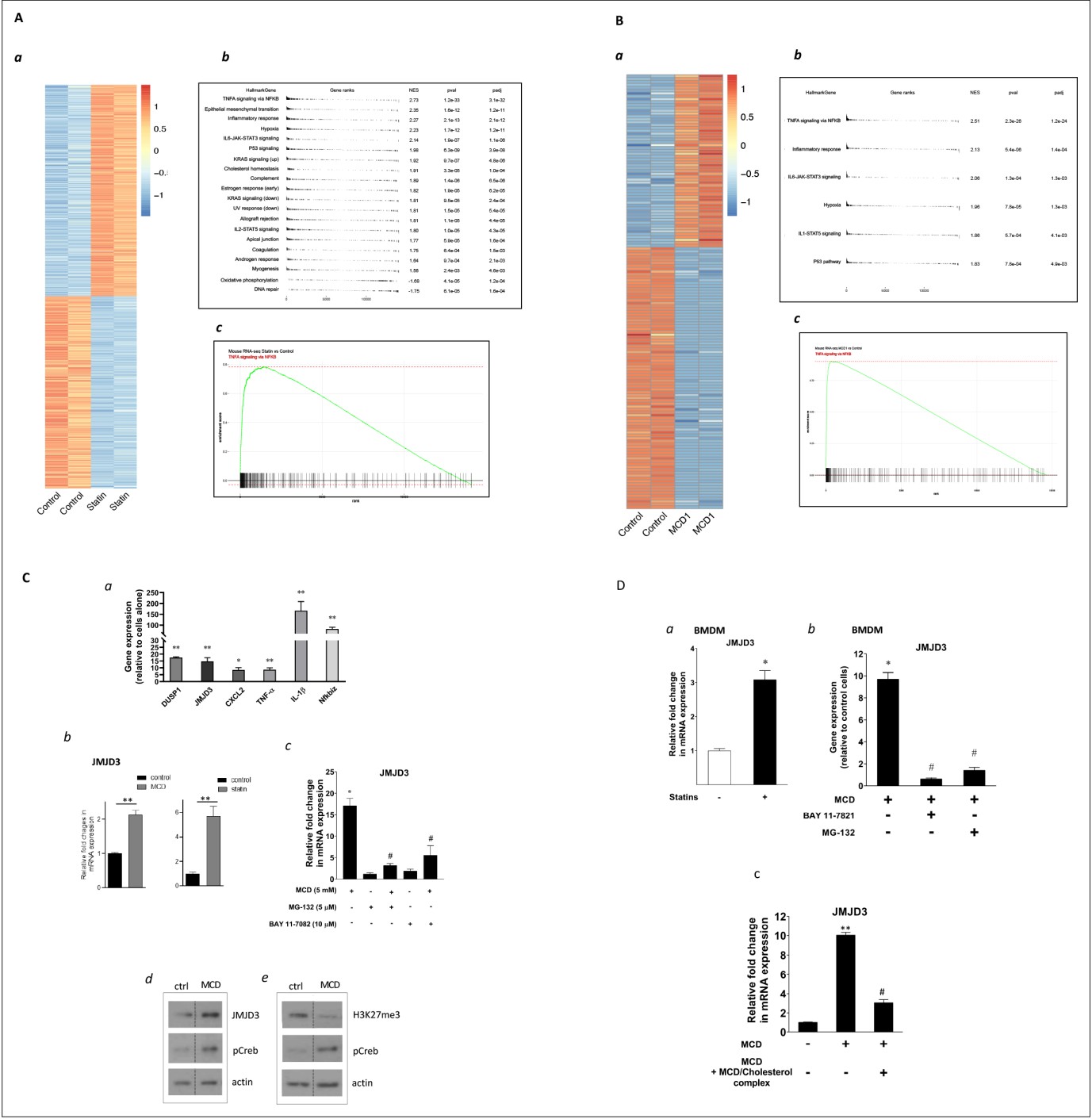

**Figure 1.** Satins upregulates the expression of *Jmjd3* in macrophages through NF-κB. (**A**) Statins activate NF-κB pathways in RAW 264.7 cells. (**a**) Heatmap of differentially expressed genes with or without statin (lovastatin, 7 + 200 μM mevalonate; 2 days). (**b**) Pathways identified by gene set enrichment analysis (GSEA) of differentially expressed genes in (**a**). (**c**) The details of most highly represented pathway, TNFA signaling vis NF-κB. (**B**) Methyl-β-cyclodextrin (MCD) activate NF-κB pathways in RAW 264.7 cells. (**a**) Heatmap of differentially expressed genes with or without MCD (5 mM, 1 hr). (**b**) Pathways identified by GSEA of differentially expressed genes in (**a**). (**c**) The details of most highly represented pathway, TNFA signaling vis NF-κB. (**C**) Statins and MCD upregulate *Jmjd3* in RAW 264.7 cells. (**a**) Reverse transcriptase quantitative PCR (RT-qPCR) of genes with or without MCD (5 mM; 1 hr). (**b**) *Jmjd3* gene expression in MCD- or statin-treated RAW 264.7 macrophages. (**c**) Effect of NF-κB inhibitors, MG-132 (5 μM) and BAY11-7082 (10 μM) on *Jmjd3* expression in MCD-treated RAW macrophages. (**d**) Western blotting of JMJD3 protein expression and (**e**) levels of H3K27Me3 in macrophages treated with 5 mM MCD (1 hr). The pCREB was used as internal control for cholesterol depletion and actin a loading control. Original blots are in source data. (**D**) Statin/MCD upregulates *Jmjd3* in bone marrow-derived macrophages (BMDMs). (**a**) *Jmjd3* gene expression in statin-treated BMDMs (10 μM pravastatin + 200 μM mevalonate; 2 days). (**b**) Effect of NF-κB inhibitors, MG-132 [5 μM] and BAY11-7082 [10 μM], on

*Figure 1 continued on next page*

*Figure 1 continued*

*Jmjd3* expression in MCD-treated BMDMs. (**c**) *Jmjd3* expression in cholesterol repletes MCD-treated macrophages. Graphs are representative of 3 independent experiments with 3 replicates per condition and are presented as means ± standard deviation (SD). Statistical analysis was performed using unpaired, two-tailed Student's *t*-test. An asterisk (*) or double asterisks (**) indicate a significant difference with p < 0.05 and p < 0.001, respectively. A hashtag (#) indicates a significant difference between MCD without or with inhibitors, p < 0.05.

The online version of this article includes the following source data and figure supplement(s) for figure 1:

**Source data 1.** Original scanned films of western blots for *Figure 1C,d*.

**Source data 2.** Original scanned films of western blots for *Figure 1C,e*.

**Figure supplement 1.** Cellular cholesterol contents regulate NF-*κ*B pathway.

**Figure supplement 2.** The expression of *Jmjd3* activated by methyl-β-cyclodextrin (MCD) requires NF-*κ*B activity.

**Figure supplement 3.** Levels of H3K27Me3 are decreased in macrophages treated with statin or methyl-β-cyclodextrin (MCD) (lovastatin, 7 + 200 µM mevalonate; 2 days) or MCD (5 mM, 1 hr).

**Figure supplement 3—source data 1.** Original scanned films of western blots for *Figure 1—figure supplement 3*.

Additional analysis, using a mouse transcriptional regulatory network database (*Han et al., 2018*; *Zhou et al., 2019*), also identified the *Nfkb1* as the top transcription factor (TF) in the promoters of genes upregulated by statins or MCD treatment (*Figure 1—figure supplement 1A, B*). Genes down-regulated by statins, on another hand, are less enriched in TFs (*Figure 1—figure supplement 1C*). In fact, MCD treatment produced no enrichment of TFs among downregulated genes. Together, the RNA-sequencing data suggest that reducing cholesterol in macrophages primarily activates NF-κB pathways to enhance gene expression.

We investigated several known NF-κB target genes using qPCR and noted that *Kdm6b*, encoding the H3K27me3 demethylase *Jmjd3* (*De Santa et al., 2007*), was among the upregulated genes (*Figure 1C, a*). Both statins and MCD enhanced *Jmjd3* expression (*Figure 1C, b*). This upregulation was abolished by several structurally unrelated NF-κB inhibitors (*Figure 1C and c* and *Figure 1—figure supplement 2*), confirming that *Jmjd3* is a NF-κB target (*De Santa et al., 2009*). We further confirmed that JMJD3 protein level is increased (*Figure 1C, d*) and that the level of H3K27me3, the substrate of JMJD3, is reciprocally decreased in MCD-treated macrophages - with reduced choles-terol (*Figure 1C, e*). The elevated level of phosphorylated Creb (pCreb) is an indicator of effective cholesterol reduction (*Ma et al., 2012*). Similar results were seen in macrophages treated with statins (*Figure 1—figure supplement 3*). In addition, mouse bone marrow-derived macrophages (BMDMs) responded to cholesterol reduction identically: *Jmjd3* is upregulated by statins or MCD in an NF-κB-dependent manner (*Figure 1D, a, b*). Lastly, to verify the specificity of cholesterol in *Jmjd3* upregu-lation, MCD/cholesterol complex was used to replenish cellular cholesterol in MCD-treated BMDMs. This reversed the *Jmjd3* upregulation by cholesterol reduction in BMDMs (*Figure 1D, c*). We conclude that statin/MCD upregulates *Jmjd3* in macrophages likely through cholesterol reduction and NF-κB activation.

## Reducing cholesterol in macrophages directly activates NF-κB

GSEA above identifies the NF-κB pathway as the top activated biological process in both statin- and MCD-treated macrophages (*Figure 1A, B*). We thus directly examined NF-κB activities in macro-phages treated with statins or MCD. Using RAW-Blue cells containing an NF-κB reporter (*Hansen et al., 2015*), we observed elevated NF-κB activity when cells were treated with MCD or statins (*Figure 2A, a, b*). In addition, two known NF-κB target genes, *Il1b* and *Tnfa*, were upregulated by MCD and statins, respectively (*Figure 2A c, d*). Furthermore, blocking NF-κB function by inhibitors prevented the upregulation of *Il1b* and *Tnfa* by MCD (*Figure 2A, e, f*). Lastly, to understand the scope of NF-κB activation by statins or MCD, we directly compared the magnitude of *Il1b* and *Tnfa* expres-sion by statins or MCD, with those by LPS. The effect of statins or MCD is much weaker than LPS: the changes in *Il1b* and *Tnfa* levels of expression by statin/MCD are less than 1% of those stimulated by LPS (*Figure 2B, a, b*). We conclude that cholesterol reduction in macrophages likely activates NF-κB. However, this activation of NF-κB is of low magnitude, and distinct from the more robust NF-κB acti-vation stimulated by LPS.

**Table 1.** Upregulated genes (top 40) by statin (A) and methyl-β-cyclodextrin (MCD) (B).

RNA-seq was performed in statin- or MCD-treated macrophages. For genes upregulated (log$_2$ >1) (*Supplementary file 1*), the top 40 upregulated genes by both statin and by MCD.

| Gene names | log$_2$ fold change | p value | p$_{adj}$ |
|---|---|---|---|
| Pgf | 10.13651906 | 8.87E−12 | 1.50E−10 |
| Rab15 | 9.393223225 | 4.73E−10 | 6.56E−09 |
| Sphk1 | 8.930573917 | 1.11E−09 | 1.45E−08 |
| Cplx2 | 8.774337587 | 1.02E−08 | 1.15E−07 |
| Edn1 | 8.701794125 | 4.44E−10 | 6.21E−09 |
| Dok7 | 8.435798979 | 0.000000194 | 1.79E−06 |
| Pdcd1 | 7.677598898 | 0.00000317 | 0.000023229 |
| Bdkrb2 | 7.485212923 | 0.00000921 | 0.000061728 |
| Pdpn | 7.245557725 | 8.11E−10 | 1.09E−08 |
| Nr4a3 | 6.905154247 | 8.81E−17 | 2.79E−15 |
| Dusp8 | 6.790646438 | 4.60E−46 | 1.27E−43 |
| Areg | 6.694617375 | 9.88E−11 | 1.48E−09 |
| Gm13889 | 6.67201557 | 1.19E−33 | 1.55E−31 |
| Ptpn14 | 6.580623988 | 9.83E−13 | 1.91E−11 |
| Hgfac | 6.329990066 | 0.0000575 | 0.00031881 |
| Clec4n | 6.157949617 | 1.52E−114 | 4.62E−111 |
| Mylk2 | 6.131939756 | 0.00163753 | 0.0060117 |
| Socs2 | 6.088256349 | 0.00180709 | 0.00656174 |
| Hr | 6.018030298 | 8.71E−42 | 2.08E−39 |
| Arntl2 | 6.016235026 | 0.00019782 | 0.00095687 |
| Esyt3 | 6.006389797 | 0.00297479 | 0.01010416 |
| Arhgef26 | 5.979707067 | 0.00654321 | 0.01983335 |
| Itgb3 | 5.924269828 | 0.00404747 | 0.01320581 |
| Tnfrsf9 | 5.761582722 | 2.79E−09 | 3.42E−08 |
| Hbegf | 5.737032919 | 1.23E−16 | 3.81E−15 |
| Sema7a | 5.586005669 | 0.00826075 | 0.02417248 |
| Lrrc32 | 5.473546382 | 0.00000044 | 3.80E−06 |
| Scn11a | 5.408935346 | 0.01371225 | 0.03712608 |
| \Prss35 | 5.402316379 | 0.01365624 | 0.0370148 |
| Mustn1 | 5.257604938 | 0.000000142 | 1.35E−06 |
| Rgs16 | 5.190735177 | 0.000000529 | 0.000004505 |
| Clcf1 | 5.151265251 | 0.00361481 | 0.01196418 |
| BC021614 | 5.075557852 | 0.00411785 | 0.01339593 |
| Cxcl2 | 5.021314952 | 9.02E−191 | 1.10E−186 |
| Fam20a | 4.966798875 | 0.00741 | 0.02203267 |
| Sp7 | 4.928788817 | 0.00615636 | 0.01883905 |
| Gprc5a | 4.927629461 | 6.27E−19 | 2.49E−17 |

*Table 1 continued on next page*

*Table 1 continued*

| Gene names | log$_2$ fold change | p value | p$_{adj}$ |
|---|---|---|---|
| Nfe2 | 4.875385857 | 1.23E−14 | 3.03E−13 |
| Ccl3 | 4.847246887 | 1.61E−119 | 6.54E−116 |
| Lamc2 | 4.837832794 | 1.68E−09 | 2.13E−08 |
| Il1b | 4.115965111 | 6.22E−29 | 1.71E−26 |
| Cxcl2 | 3.962459756 | 0 | 0 |
| Egr1 | 3.365911716 | 2.49E−125 | 2.10E−122 |
| Tnf | 3.306387647 | 1.50E−176 | 2.06E−173 |
| Nfkbiz | 3.259909052 | 0 | 0 |
| Arc | 3.021670065 | 2.13E−30 | 6.18E−28 |
| Ier3 | 2.897093474 | 4.02E−153 | 4.43E−150 |
| Zfp36 | 2.853262721 | 1.29E−157 | 1.57E−154 |
| Lif | 2.748205257 | 1.80E−10 | 2.18E−08 |
| Dusp2 | 2.627829764 | 7.14E−129 | 6.55E−126 |
| Tnfaip3 | 2.585932017 | 2.15E−279 | 7.87E−276 |
| Egr2 | 2.32626986 | 2.58E−46 | 9.46E−44 |
| Ier2 | 2.034656801 | 2.63E−262 | 7.24E−259 |
| Dusp1 | 1.977596852 | 6.88E−109 | 4.45E−106 |
| Junb | 1.950358114 | 5.59E−74 | 2.80E−71 |
| Btg2 | 1.92138967 | 3.61E−204 | 7.94E−201 |
| Pde4b | 1.920348453 | 8.18E−87 | 4.50E−84 |
| Dusp5 | 1.918194267 | 6.50E−27 | 1.75E−24 |
| Maff | 1.891310296 | 8.36E−41 | 2.70E−38 |
| Ppp1r15a | 1.878276549 | 6.16E−121 | 4.84E−118 |
| Ptgs2 | 1.775605515 | 1.44E−201 | 2.63E−198 |
| Phlda1 | 1.775056479 | 1.24E−13 | 1.90E−11 |
| Nfkbid | 1.673557595 | 4.68E−111 | 3.22E−108 |
| Fos | 1.629250783 | 1.79E−146 | 1.79E−143 |
| Nfkbia | 1.595064568 | 7.69E−196 | 1.21E−192 |
| Nr4a1 | 1.580570737 | 3.42E−42 | 1.18E−39 |
| Socs3 | 1.571871776 | 9.72E−66 | 4.11E−63 |
| Gdf15 | 1.555447107 | 0.0000514 | 0.00317852 |
| Pim1 | 1.547649464 | 5.43E−108 | 3.32E−105 |
| Zc3h12a | 1.524253928 | 7.40E−92 | 4.28E−89 |
| Egr3 | 1.494163502 | 5.44E−20 | 1.13E−17 |
| Osm | 1.430409511 | 1.27E−22 | 3.10E−20 |
| Myc | 1.331901297 | 1.10E−10 | 1.35E−08 |
| Gpr84 | 1.310067562 | 1.23E−18 | 2.42E−16 |
| Traf1 | 1.30684728 | 4.76E−52 | 1.81E−49 |
| Cxcl10 | 1.28466895 | 1.01E−15 | 1.76E−13 |

*Table 1 continued on next page*

*Table 1 continued*

| Gene names | log₂ fold change | p value | p_adj |
|---|---|---|---|
| Hmgcs1 | 1.279756645 | 7.39E−07 | 6.25E−05 |
| Errfi1 | 1.254057102 | 1.47E−38 | 4.63E−36 |
| H2-K2 | 1.251411578 | 1.71E−17 | 3.24E−15 |
| Ccrl2 | 1.214065207 | 1.41E−13 | 2.13E−11 |

## Reducing cholesterol in macrophages decreases mitochondria respiration

We next explored potential mechanisms by which cholesterol reduction with statin/MCD activates NF-κB. In recent years, it has become evident that NF-κB can be activated by a metabolic shift in the cell from oxidative phosphorylation (OXPHOS) in the mitochondria to glycolysis in the cytoplasm (*Tannahill et al., 2013*; *Jha et al., 2015*). We therefore investigated whether statin/MCD modulates OXPHOS in macrophages to activate NF-κB. Using the extracellular flux analyzer (Seahorse), we found that 1 hr MCD treatment of BMDMs decreased overall resting mitochondrial oxygen consumption rate (OCR) (*Figure 3A, a*). Moreover, this decrease was entirely attributable to the suppression of the ATP synthase (*Figure 3A, b*), a protein embedded in the inner mitochondrial membrane (IMM) and a component of electron transport chain. Interestingly, the maximal respiration remains unchanged by MCD (*Figure 3A, c*). Statin treatment similarly decreased overall resting mitochondrial OCR (*Figure 3B, a*) and that of ATP synthase (*Figure 3B, b*). However, statins also lowered the maximal respiration (*Figure 3B, c*), possibly due to the 2-day treatment period required for the statin treatment. Overall, statin or MCD suppresses ATP synthase in macrophages.

## Reducing cholesterol in macrophages results in lower cholesterol level in IMM, which suppresses ATP synthase activity

We next asked how cholesterol reduction by statin/MCD might influence the ATP synthase in the IMM. In mammalian cells, the majority of cholesterol is in the plasma membrane. However, all intracellular membranes, including those in the mitochondria, also contain cholesterol (*Steck and Lange, 2010*). Cholesterol distribution among cellular membranes is governed by a dynamic steady state (*Lange and Steck, 2020*), such that reduction in total cellular cholesterol content by statin or MCD will decrease cholesterol levels in all membranes, including those in the mitochondria (*Lange et al., 2009*). Levels of IMM cholesterol can be assessed by the activity of an IMM enzyme, sterol 27-hydroxylase (CYP27A1), which catalyzes the conversion of cholesterol to 27-hydroxycholesterol (27-HC) as a function of cholesterol availability in IMM (*Li et al., 2006*). The amount of 27-HC in macrophages thus directly reflects IMM cholesterol levels, if CYP27A1 remains constant (*Lange et al., 2009*). We found that *Cyp27a1* levels are significantly reduced by 2-day statin treatment (*Figure 4—figure supplement 1*) but remain steady after 1-hr MCD treatment (*Figure 4A*). We thus analyzed 27-HC contents by mass spectrometry on MCD-treated macrophages to assess the cholesterol levels in IMM. We found that the amount of cellular 27-HC was decreased in a dose-dependent manner after a 1-hr MCD treatment (*Figure 4B*), indicating a reduction in cholesterol levels in IMM by MCD. Statins decrease total cholesterol content in macrophages similar to MCD (*Ma et al., 2012*), which should then similarly lower cholesterol levels in IMM, even though this particularly assay could not be applied due to the changes in *Cyp27a1* expression described above.

To understand precisely how cholesterol levels in the IMM might influence ATP synthase functions, we used an in vitro membrane system composed of mitochondria inner membrane vesicles (IMVs) from *Escherichia coli* (*van der Does et al., 2003*). *E. coli* inner membrane shares common features with those from mammalian cells (*Meyer et al., 2018*), including ATP synthase functions (*Nirody et al., 2020*), but lacks cholesterol or any other sterol derivative in their lipid composition (*Sohlenkamp and Geiger, 2016*). Various levels of cholesterol can be incorporated into the IMVs to assess the impact of cholesterol (*Mahammad and Parmryd, 2015b*). With this system, we observed that the activity of the ATP synthase, both in synthesis and hydrolysis mode, is highly sensitive to cholesterol concentration in the membrane: the highest activity is found in IMV with membrane containing 7% cholesterol; decreasing

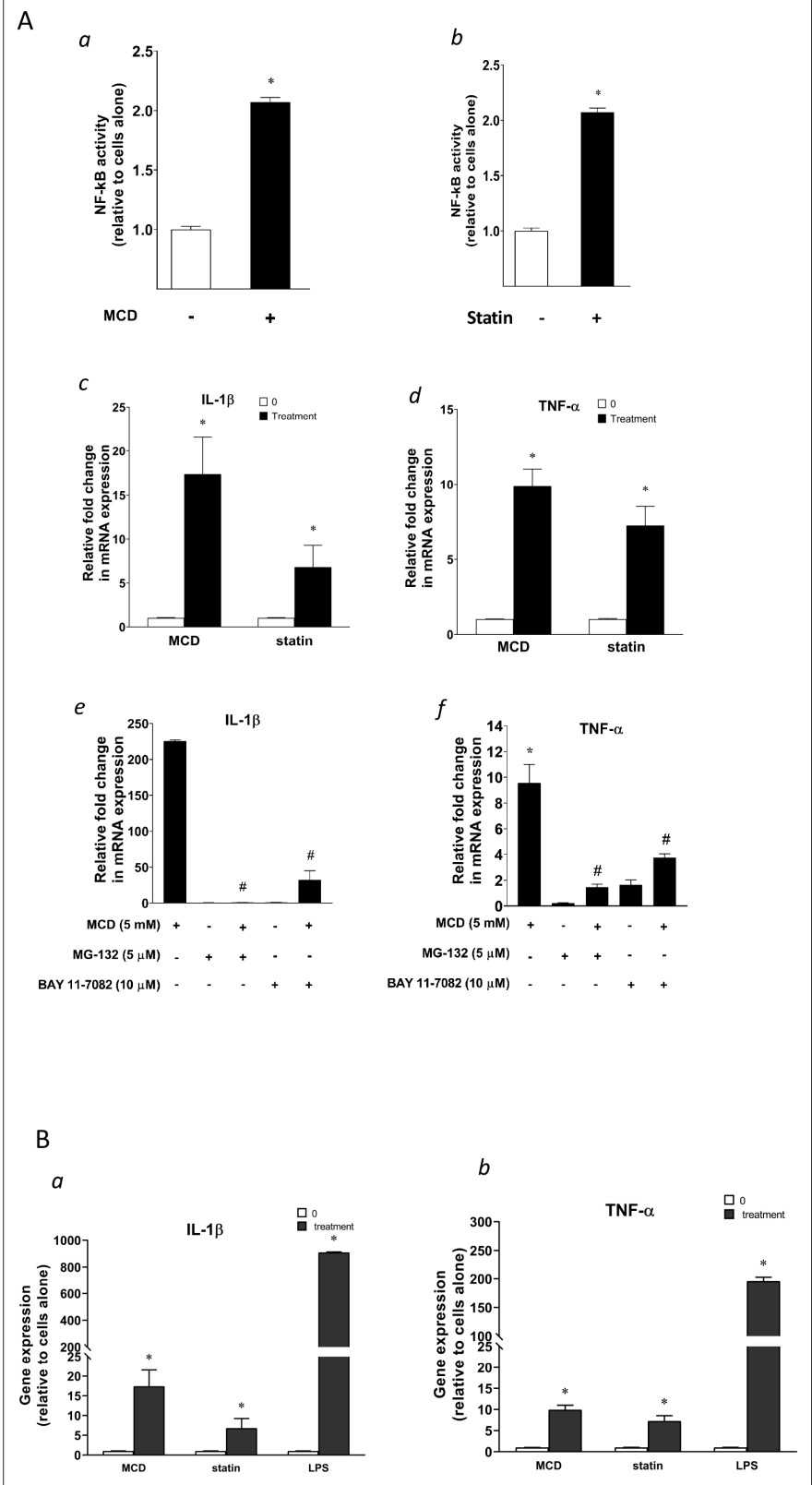

**Figure 2.** Cellular cholesterol contents regulate NF-κB pathway. (**A**) NF-κB activation by methyl-β-cyclodextrin (MCD) (**a**) and statins (**b**) using RAW blue macrophages. Reverse transcriptase quantitative PCR (RT-qPCR) of *Il1b* (**c**) and *Tnfa* (**d**) in RAW 264.7 cells with or without MCD (5 mM; 1 hr), or with or without (10 μM compactin + 200 μM mevalonate; 2 days). Effect of NF-κB inhibitors (MG-132 [5 μM] and BAY11-7082 [10 μM]) on *Il1b* (**e**) and

*Figure 2 continued on next page*

*Figure 2 continued*

*Tnfa* (**f**) expression in MCD-treated RAW 264.7 macrophages. (**B**) The gene expression activated by MCD, statin, or lipopolysaccharide (LPS). RT-qPCR of *Il1b* (**a**) and *Tnfa* (**b**) in RAW 264.7 cells with or without MCD (5 mM; 1 hr), simvastatin (10 + 200 µM mevalonate; 2 days) or LPS (100 ng/ml; 3 hr). Graphs are representative of 3 independent experiments with 3 replicates per condition and are presented as means ± standard deviation (SD). Statistical analysis was performed using unpaired, two-tailed Student's *t*-test. An asterisk (*) indicates a significant difference with $p < 0.05$. A hashtag (#) indicates a significant difference between MCD without or with inhibitors with $p < 0.05$.

cholesterol from 7% cholesterol suppresses ATP synthase activities (*Figure 4C*). The steady-state IMM cholesterol level in mammalian cells is about 5% (*Horvath and Daum, 2013*). If MCD reduces cholesterol in IMM, as we show above (*Figure 4B*), ATP synthase activity should be suppressed, as we have seen in OCR (*Figure 3A, b*). Therefore, the in vitro experimental model is consistent with the notion that MCD lowers cholesterol in the IMM, which suppresses ATP synthase activity.

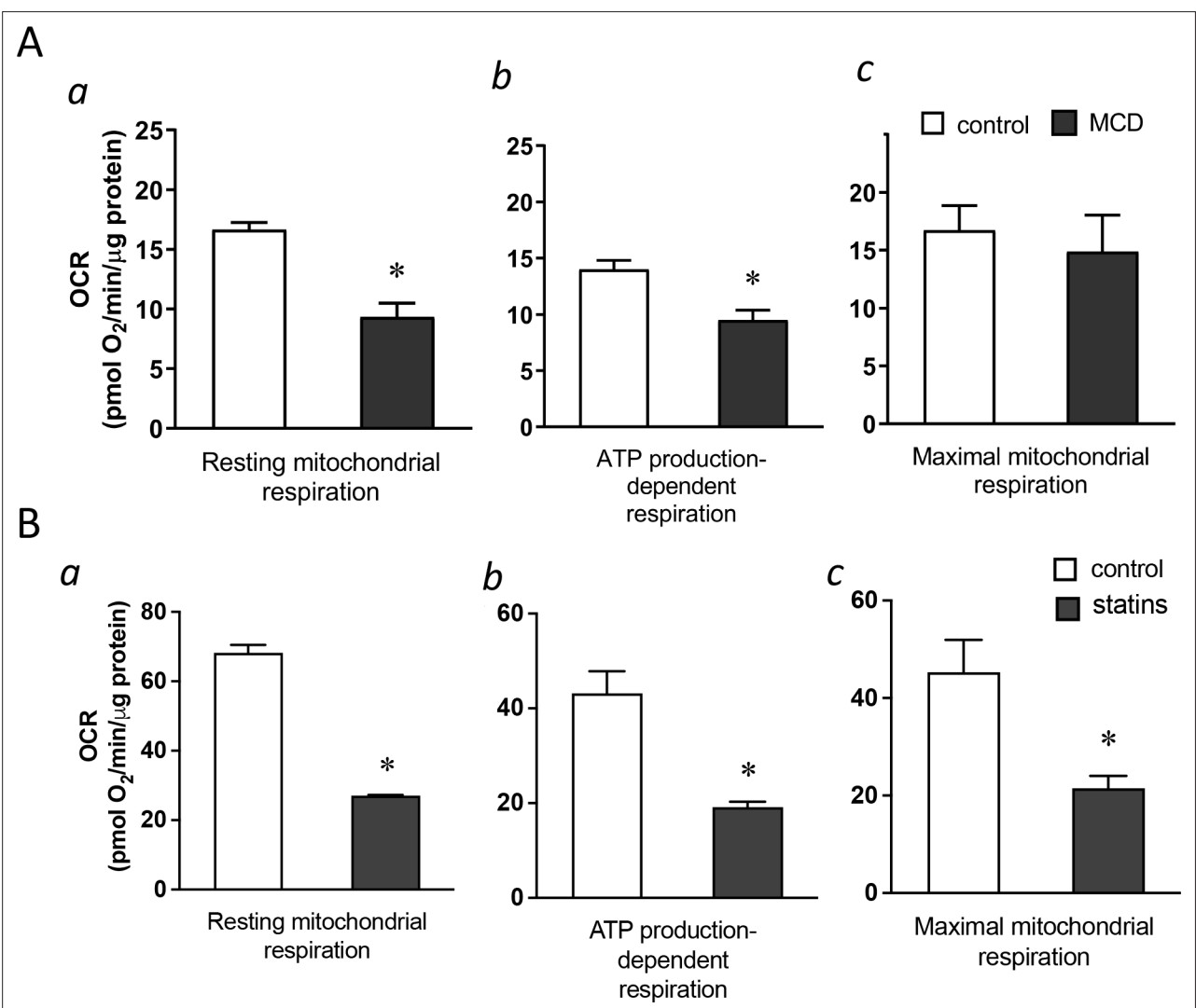

**Figure 3.** Cholesterol levels modulate mitochondrial respiration. (**A**) Mitochondrial oxygen consumption rates (OCRs) of bone marrow-derived macrophages (BMDMs) treated with 1 hr methyl-β-cyclodextrin (MCD) (5 mM) or without. (**a**) OCR for mitochondrial resting respiration. (**b**) OCR representing mitochondrial ATP production-linked respiration. (**c**) OCR representing maximal respiration. (**B**) Mitochondrial OCRs in BMDMs treated compactin or without compactin (10 + 200 µM mevalonate; 2 days). (**a**) OCR for mitochondrial resting respiration. (**b**) OCR representing mitochondrial ATP production-linked respiration. (**c**) OCR representing maximal respiration. Data are representative of 3 independent experiments with 3 samples per group and data are presented as mean ± standard deviation (SD). Statistical analysis was performed using unpaired, two-tailed Student's *t*-test. An asterisk (*) indicates a significant difference, $p < 0.05$.

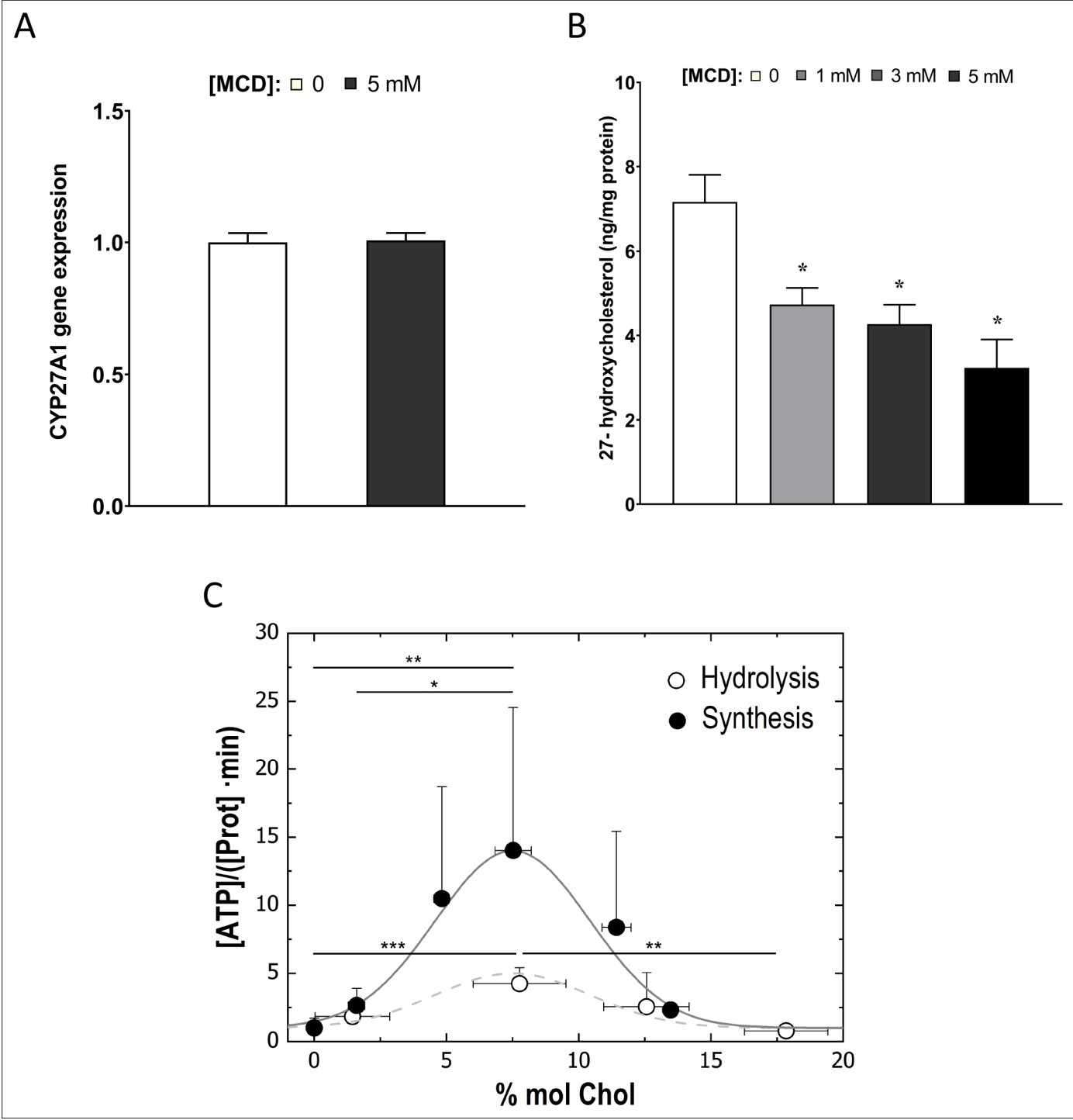

**Figure 4.** Cholesterol levels modulate the mitochondrial ATP synthase activity. (**A**) Sterol 27-hydroxylase, *Cyp27a1*, expression in 1 hr methyl-β-cyclodextrin (MCD)-treated and control RAW macrophages. (**B**) 27-Hydroxycholesterol (27-HC) analysis by ultraperformance liquid chromatography/electrospray ionization/tandem mass spectrometry. 27-HC levels were normalized to the protein content in the whole cell pellet. (**C**) ATP hydrolysis and synthesis in cholesterol-doped inner membrane vesicles (IMVs); ATP hydrolysis was performed by adding a total concentration of 2 mM ATP to 200 mM IMVs (lipid concentration) and incubated for 30 min. Concentration of phosphates from ATP hydrolysis was measured using the malaquite green assay. ATP concentration after synthesis was measured using ATP detection assay kit (Molecular Probes) with a luminometer GloMax-Multi Detection. Data in B are from 3 samples per group and data are presented as mean ± standard deviation (SD). An asterisk (*) indicates p < 0.05. Data in (**C**) are representative of at least two independent experiments with three replicates and presented as means ± SD. Statistical analysis was performed using the Tukey analysis of variance (ANOVA) test. (*), (**), and (***) indicate a significant difference with p < 0.05, 0.01, and 0.001, respectively.

*Figure 4 continued on next page*

*Figure 4 continued*

The online version of this article includes the following figure supplement(s) for figure 4:

**Figure supplement 1.** Sterol 27-hydroxylase, Cyp27a1, expression.

## Reducing cholesterol levels in macrophages alters proton gradients in the mitochondria to activate NF-κB and upregulate Jmjd3

We next studied the potential role of suppressed ATP synthase on NF-κB activation and *Jmjd3* upregulation. ATP synthase in the IMM uses proton flow from the inner space to the matrix to generate ATP (*Figure 5A, a*). As shown by extracellular flux analysis (*Figure 3*), MCD suppresses the activity of ATP synthase. This will lead to fewer protons flowing down from the inner space into the matrix and, consequently, more protons will be retained in the inner space in MCD-treated macrophages (*Figure 5A, b*), which could activate NF-κB (*Mills et al., 2016*). We tested this using carbonyl cyanide *m*-chlorophenyl hydrazine (CCCP), a mitochondrial proton ionophore that prevents proton buildup in the inner space (*Lou et al., 2007*; *Figure 5A, c*). Indeed, in the presence of CCCP, MCD failed to activate NF-κB (*Figure 5B, a*). CCCP also prevented MCD from upregulating IL-1β (*Figure 5B, b*). In addition, it abolished *Jmjd3* upregulation by MCD in RAW macrophages and BMDMs (*Figure 5C, a, b*), while cells remained fully viable (*Figure 5—figure supplement 1A*). Moreover, another structurally unrelated mitochondrion proton ionophore BAM15 (*Kenwood et al., 2014*) similarly abolished MCD-induced *Il1b* and *Jmjd3* expression (*Figure 5—figure supplement 1a, b*). We conclude that reducing cholesterol in macrophages suppresses ATP synthetase activity in the IMM, which likely activates NF-κB and upregulates *Jmjd3*.

## Reducing cholesterol in macrophages alters chromatin structure

The NF-κB target gene *Jmjd3* primarily functions to demethylate H3K27me3, an abundant epigenetic mark associated with transcriptional repression (*Klose et al., 2006*). The upregulation of *Jmjd3* by statin/MCD is expected to decrease H3K27me3 levels, which should have an impact on the macrophage epigenome. We performed the assay for transposase-accessible chromatin with sequencing (ATAC-seq) to compare the transposase accessibility with or without MCD. We observed that MCD treatment significantly altered the genomic locations of open/close chromatin in macrophages (*Figure 6A, a* and *Supplementary file 1*). Consistent with our RNA-seq studies, GSEA of all genes showing altered chromatin accessibility upon MCD treatment identified NF-κB pathway as the top biological process (*Figure 6A, b, c*). We also analyzed genes being opened by MCD: they predominantly have NF-κB family of TFs in promoters (*Figure 6—figure supplement 1*). We then compared ATAC-seq with RNA-seq and identified overlaps genes, i.e., increased accessibility to transposase and higher expression (*Supplementary file 4*). Noticeably, *Jmjd3*, *Il1b*, and *Tnfa* are among those. Also, *Jmjd3* is the only gene with epigenetic modification function. The details of *Il1b* and *Tnfa* are shown in *Figure 6B*. We conclude that the epigenome is altered by cholesterol reduction in macrophages.

## Reducing cholesterol promotes anti-inflammatory responses in activated macrophages

The reconfigured epigenome upon cholesterol depletion could poise resting macrophages for different inflammatory responses. We therefore tested inflammatory responses in statin-treated macrophages with LPS (classically activated or M1 phenotype) or with IL-4 (alternatively activated or M2 phenotype). Macrophages were treated with statins or without and then stimulated by LPS. RNA-seq was performed. Cholesterol reduction by statins altered a large number of genes, up or down, in LPS-stimulated macrophages (*Figure 7A, a*; *Supplementary file 5*). Gene Ontology (GO) analysis of genes with decreased expression upon statin treatment revealed that statins primarily suppress inflammatory processes (*Figure 7A, b*), while genes involved in cellular homeostatic functions were upregulated (*Figure 7A, c*). More specifically, in BMDMs activated by LPS, statin treatment led to suppressed expression of proinflammatory cytokines (*Il1b*, *Tnfa*, *Il6*, and *Il12*), but enhanced expression of anti-inflammatory cytokine *Il10* (*Figure 7B*). When BMDMs were activated alternatively by IL-4, statin treatment promoted the expression of IL-4 target genes *Arg1*, *Ym1*, and *Mrc1* (*Figure 7C*). The enhanced expression of the anti-inflammatory genes (*Arg1*, *Ym1*, *Mrc1*, and *Il10*) by statins in activated macrophages, both M1 and M2, was also correlated with the removal of H3K27me3, a

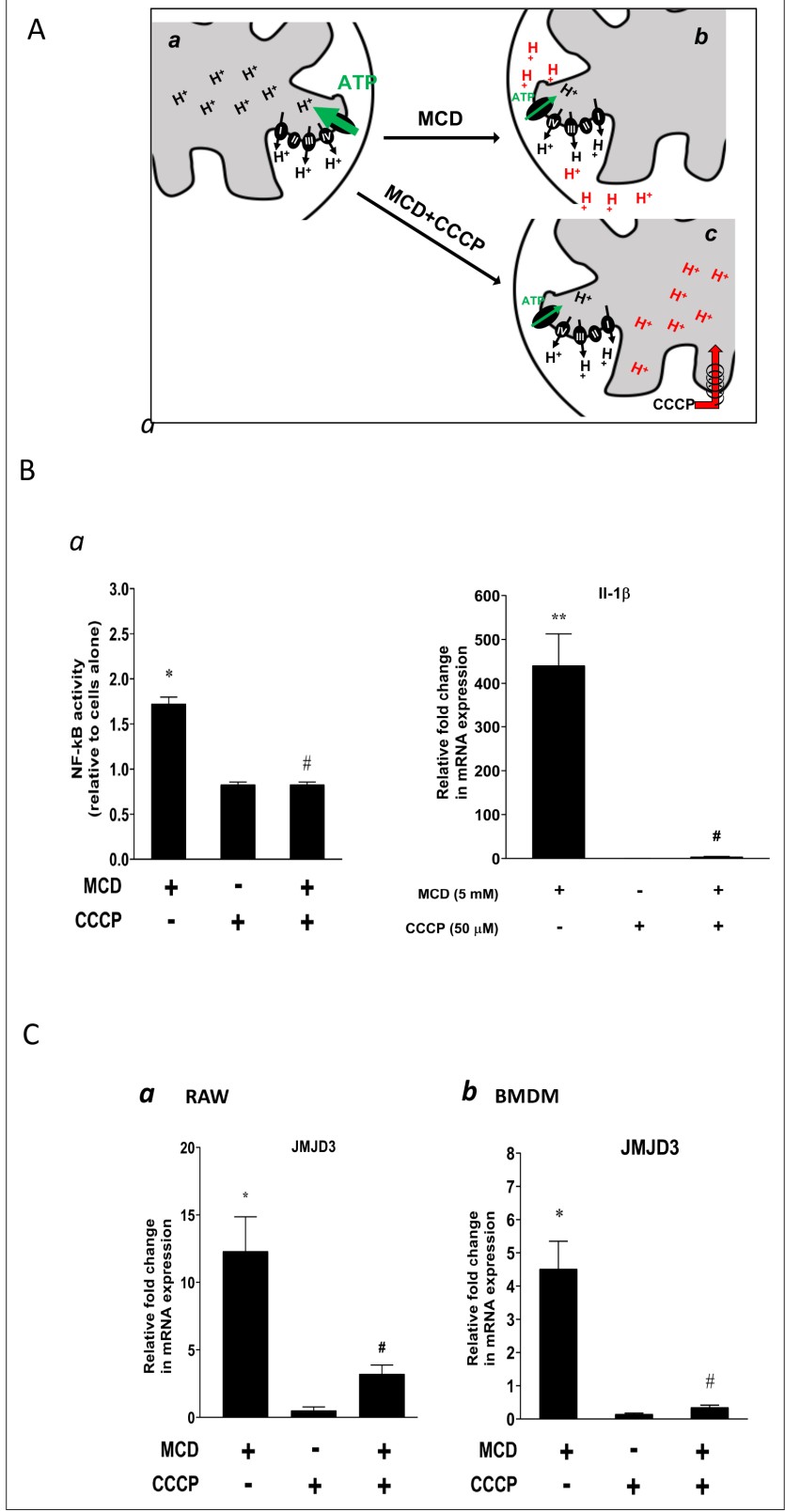

**Figure 5.** Effect of proton flux on NF-κB activation and *Jmjd3* expression in methyl-β-cyclodextrin (MCD)-treated cells. (**A**) Schematic of potential mechanism induced by MCD, and MCD/carbonyl cyanide *m*-chlorophenyl hydrazine (CCCP) on mitochondrial proton flux. (**B**) Effect of CCCP on NF-κB activity and *Jmjd3* expression: (**a**) on NF-κB activity in RAW blue macrophages (CCCP = 50 μM); (**b**) effect of CCCP on *Il1b* expression in RAW 264.7

*Figure 5 continued on next page*

*Figure 5 continued*

macrophages (CCCP = 50 µM). (**C**) Effect of CCCP (**a**) on *Jmjd3* expression in RAW 264.7 macrophages (CCCP = 50 µM) and (**b**) on bone marrow-derived macrophages (BMDMs) (CCCP = 200 µM). Data are representative of 3 independent experiments with 3 samples per group and data are presented as mean ± standard deviation (SD). Statistical analysis was performed using unpaired, two-tailed Student's *t*-test. Asterisks (*) and (**) indicate a significant difference with p < 0.05 and p < 0.001. A hashtag (#) indicates a significant difference between MCD without or with inhibitors with p < 0.05.

The online version of this article includes the following figure supplement(s) for figure 5:

**Figure supplement 1.** The effect of proton flux Inhibitors.

repressive marker and the substrate of JMJD3 in resting macrophages (*Figure 7D*). Thus, experimental evidence supports the notion that cholesterol reduction by statins in macrophages leads to less proinflammatory responses to LPS, but higher expression of anti-inflammatory genes, such as these activated by IL-4, correlated with H3K27me3 removal (*Liu et al., 2017*).

## Anti-inflammatory responses by cholesterol reduction in macrophages rely on Jmjd3 and its demethylase activities

Jmjd3 belongs to the JmjC demethylase family that requires α-ketoglutarate (α-KG) as co-factor to demethylate histone (*Liu et al., 2017*). Noticeably, *Jmjd3* (*Kdm6b*) is the only gene member of the JmjC family upregulated by MCD in macrophages and, importantly, the expression of closely related *Utx* (*Kdm6a*) is not changed by cholesterol reduction (*Figure 8A, a*). This presented an opportunity to specifically probe the involvement of *Jmjd3* demethylation activity in suppressing *Tnfa* yet raising *Il10* expression by MCD. *Jmjd3* expression is increased with MCD in a concentration-dependent manner (*Figure 8—figure supplement 1*). When subsequently stimulated by LPS, MCD-treated macrophages dose-dependently express less *Tnfa*, but more *Il10*: the ratio of *Il10/Tnfa* rises with MCD concentrations (*Figure 8A, b*, white bars). However, if glutamine, the precursor of α-KG (*Liu et al., 2017*), is absent in the medium, there is little change in the ratio of *Il10/Tnfa* (*Figure 8A, b*, black bars), regardless of MCD concentration. Furthermore, if the demethylase activity of Jmjd3 is inhibited by a specific inhibitor GSKj4 (*Kruidenier et al., 2012*), the ratio of *Il10/Tnfa* also failed to rise upon MCD treatment (*Figure 8A, c*). BMDMs similarly modify their response upon glutamine availability. When stimulated by LPS, MCD-treated BMDMs rise *Il10/Tnfa* ratio, dependent on glutamine: *Il10/Tnfa* fails to increase by MCD in the absence of glutamine (*Figure 8B, a*). Glutamine is also necessary for MCD to boost IL-4-targeted *Arg1* (*Figure 8B, b*). We next used shRNA to knockdown *Jmjd3* (*Jmjd3* KD) (*Figure 8—figure supplement 2*) and tested its impact on statin-treated macrophages. When activated by LPS, statin-treated *Jmjd3* KD macrophage failed to raise *Il10/Tnfa* ratio (*Figure 8C, a*). *Jmkd3* KD also abolished the rise of *Arg1* by statin when stimulated by IL-4 (*Figure 8C, b*). Together, we conclude that *Jmjd3* and its demethylase activity are necessary to promote the expression of anti-inflammatory elements upon cholesterol in macrophages.

## Statin treatment vivo also reduces cholesterol content, upregulates Jmjd3, and promotes anti inflammatory gene expression in macrophages

To test the in vivo effect of statins on macrophages, mice were fed with statins or not for 14 days and the peritoneal macrophages were isolated (*Becker et al., 2010*) and tested for inflammatory responses. The cholesterol content was decreased by about 20% in freshly isolated peritoneal macrophages from statin-fed mice, compared to those from control animals (*Figure 9A*). Expression of *Jmjd3* was upregulated by statin feeding (*Figure 9B*). When subsequently challenged by LPS, macrophages from statin-fed mice showed lower expression of proinflammatory cytokines (*Il1b*, *Tnfa*, *Il6*, and *Il12*), and enhanced expression of anti-inflammatory cytokine *Il10*, relative to those from controls (*Figure 9C*). In addition, when activated alternatively by IL-4, macrophages from statin-fed mice expressed higher levels of Arg*1*, *Ym1*, and *Mrc1*, compared to these from controls (*Figure 9D*). Thus, statin treatment in vivo decreases cholesterol content, upregulates *Jmjd3*, and promotes anti-inflammatory functions in freshly isolated peritoneal macrophages, in an identical fashion to that of BMDMs treated in vitro with statins (*Figure 7B, C*).

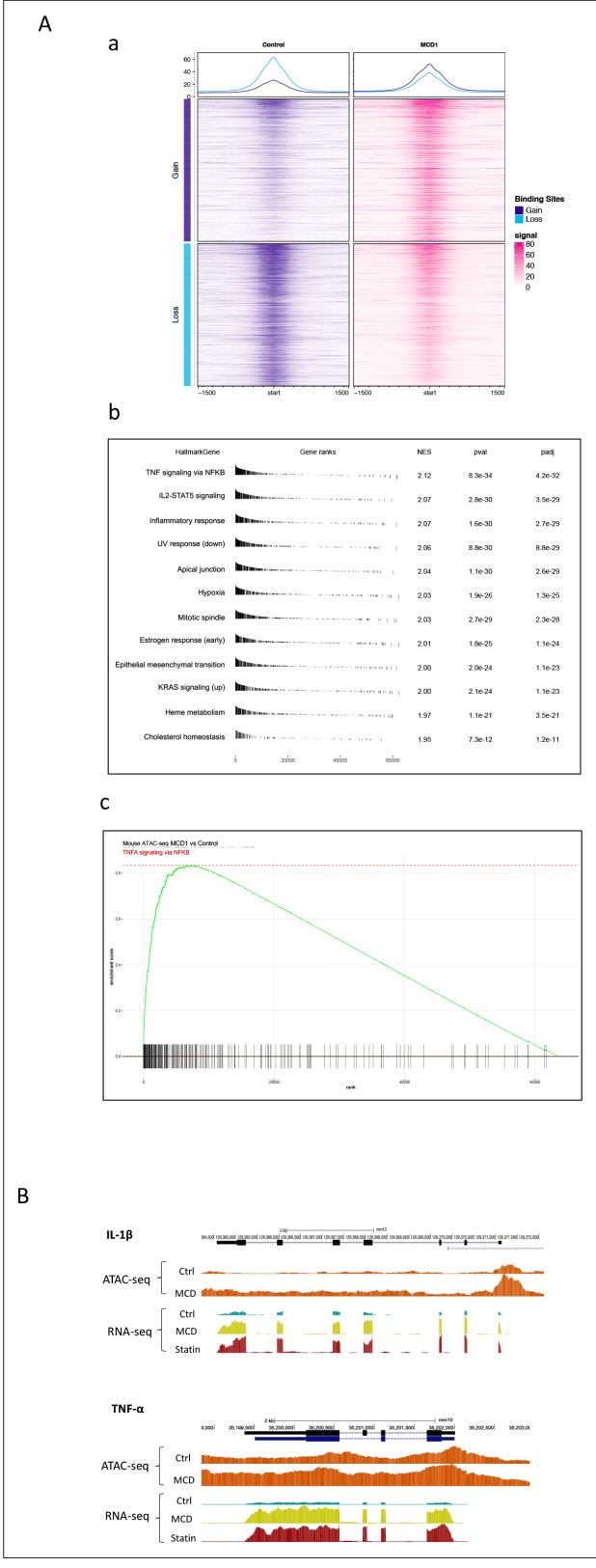

**Figure 6.** Cholesterol modulates macrophage epigenetic modifications. (**A**) ATAC-seq in control and methyl-β-cyclodextrin (MCD) (5 mM, 1 hr) treated RAW 264.7 macrophages. (**a**) Summit-centered heatmap of differentially accessible ATAC-seq signals. (**b**) Pathways identified by gene set enrichment analysis (GSEA) of differentially

*Figure 6 continued on next page*

*Figure 6 continued*

assessable genes in (**a**). (**c**) The details of most highly represented pathway, TNFA signaling vis NF- $\kappa$ B. (**B**) ATAC- and RNA-seq profiles alignment from ATAC- and RNA-seq for the genomic loci of *Il1b* (top) and *Tnfa* (bottom).

The online version of this article includes the following figure supplement(s) for figure 6:

**Figure supplement 1.** Pathway analysis among the genes opened by methyl-β-cyclodextrin (MCD) in ATAC-seq.

## Discussion

In this study, we show that reducing cholesterol in macrophages, either with statins or MCD, upregulates Jmjd3 through suppressing mitochondria respiration. Cholesterol reduction also modifies the epigenome in macrophages. Upon subsequent activation by either M1 or M2 stimuli, statin-treated macrophages are phenotypically more anti-inflammatory. This anti-inflammatory phenotype is also observed in peritoneal macrophages freshly isolated from statin-treated mice.

We speculate that *Jmid3* is the key responsible factor. Macrophages have two H3K27me3 demethylases, *Utx* and *Jmjd3*. Only *Jmjd3* is upregulated by statin or MCD. It is plausible that *Jmjd3*, by removing H3K27me3 from *Il10*, *Arg1*, *Yam1*, and *Mrc1*, poises these genes in resting macrophages to promote their activated expression. On the other hand, statins suppress a large group of proinflammatory genes activated by LPS, which could not directly be attributed to *Jmjd3*. However, *Il10* is poised by Jmjd3 and shows higher expression upon LPS stimulation. It could be that upregulation of *Il10* leads to the suppression of the proinflammatory phenotype under LPS activation. Recent studies have shown that endogenously produced IL-10, through autocrine/paracrine mechanisms, modulates mitochondria respiration to inhibit cellular events, such as glycolysis and mTORC1. This suppresses the expression of proinflammatory genes (*Ip et al., 2017*; *Dowling et al., 2021*). The elevated *Il10* expression in statin-/MCD-treated macrophages could play a similar role to suppress the proinflammatory phenotypes. Of note, this current study focuses mostly on the regulation of gene expression. The ultimate impact of statins on inflammation should be confirmed at the protein level in future studies.

We also document that reducing the level of macrophage cholesterol alters the epigenome, concurrent with *Jmjd3* upregulation. Removing H3K27me3 by JMJD3 could open certain genome regions and contribute to the changes in the epigenome seen in ATAC-seq. However, the changes we observed are much more profound, indicating that other epigenetic modifiers, are activated by cholesterol reduction. Future studies will be required to identify these modifiers. Nevertheless, several lines of evidence support the notion that *Jmjd3* is most relevant to inflammatory activation. First of all, only *Jmjd3*, among the members of the JmjC demethylase family, is upregulated by statin or MCD. *Utx*, also a H3K27me3 demethylase, is not altered by statin or MCD. Second, the JmjC demethylase family members require glutamine, the source of alpha-ketoglutarate, for demethylation. Since *Jmjd3* is the only upregulated demethylase, glutamine could be used to specifically probe *Jmjd3* demethylation function in macrophages M1/M2 activation. Glutamine is indeed necessary for MCD to modulate both LPS and IL-4 activation. Third, GSKj4, the inhibitor for H3K27me3 demethylases (Utx and Jmjd3), abolishes the MCD effect. Furthermore, knockdown of *Jmjd3* by shRNA diminishes the effect of statins on LPS or IL-4 activations. Thus, the upregulation of *Jmjd3* by statin/MCD significantly contributes to poise the epigenome in resting macrophages, which controls the subsequent inflammatory response to LPS or IL-4.

Our results here support the notion that the epigenome in resting macrophages is largely poised for inflammatory activation (*Escoubet-Lozach et al., 2011*). This also agrees with previous studies where cholesterol reduction was shown to decrease proinflammatory responses to multiple stimuli against multiple TLRs (*Yvan-Charvet et al., 2008*). We have focused on classically activated (M1) and alternatively activated (M2) phenotypes, two extremes of inflammatory activation in macrophages. This could be an experimental starting point, since macrophages likely encounter a wide range of stimuli within a continuum between M1 and M2 phenotypes. The poised epigenome in resting macrophages, i.e., prior to activation, may serve as an initial blueprint to propel the activated inflammatory responses. In addition, the inflammation processes are thought to initially engage the M1 phenotype to fight pathogens and then gradually gain the M2 phenotype to restrain excessive damage and restore tissue homeostasis (*Ivashkiv, 2013*).

Another novel finding here is that the level of cellular cholesterol directly controls mitochondria respiration. Except for specialized cell types (i.e., steroidogenic cells), the mitochondrial cholesterol in

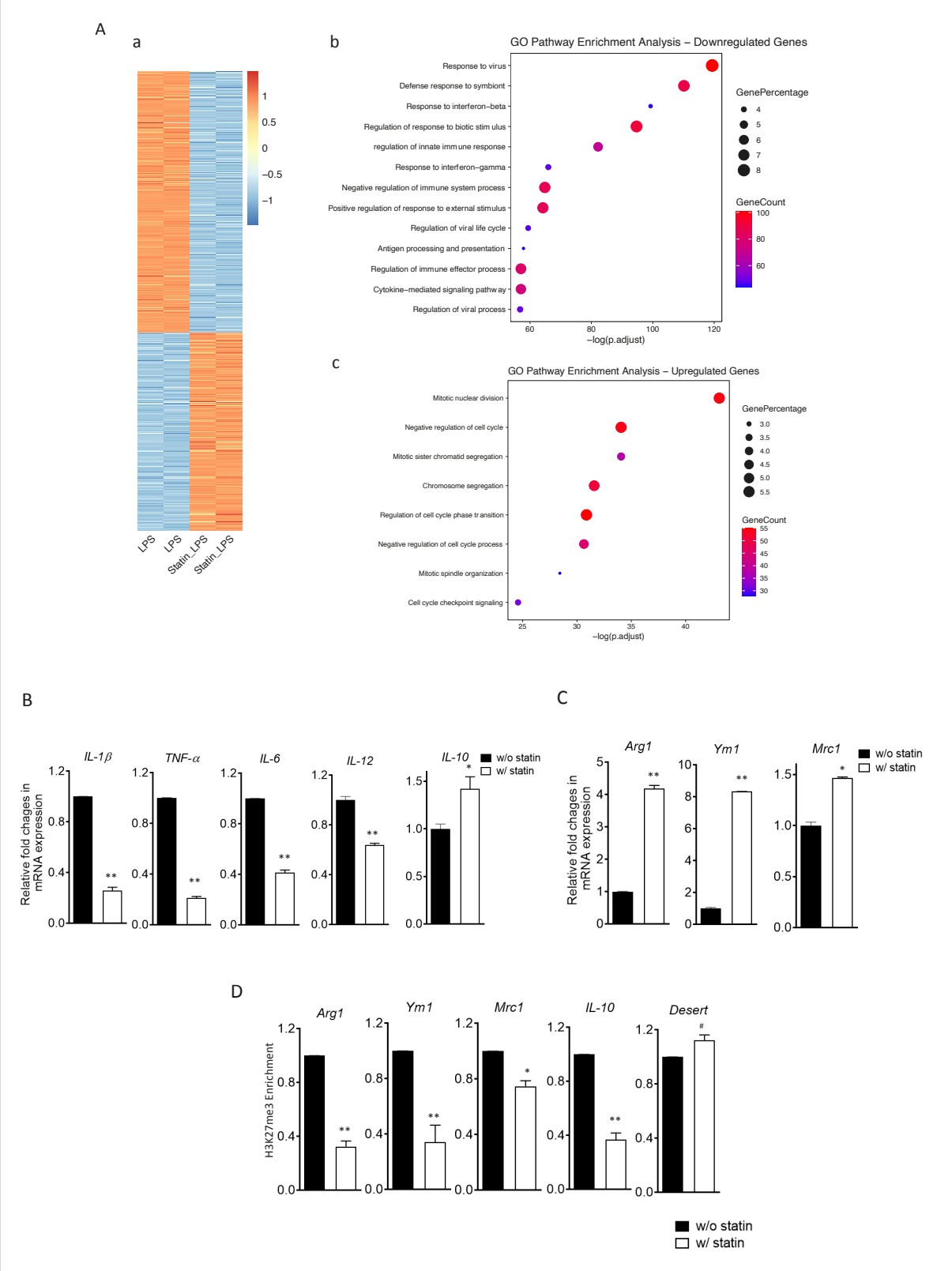

**Figure 7.** Statins supress proinflammatory cytokines and enhance anti-inflammatory factors in lipopolysaccharide (LPS)- or IL-4-activated macrophages. (**A**) RAW 264.7 macrophages treated with statins (lovastatin, 7 + 200 µM mevalonate; 2 days) or without were stimulated with LPS (100 ng/ml) for 3 hr. (**a**) Heatmap of differentially expressed genes by statins. (**b**) Gene Ontology (GO) analysis of downregulated genes by statins. (**c**) GO analysis of upregulated genes by statin. (**B**) Bone marrow-derived macrophages (BMDMs) treated with or without compactin (10 + 200 µM mevalonate; 2 days)

*Figure 7 continued on next page*

*Figure 7 continued*

are stimulated by LPS (50 ng/ml, 3 hr). Gene expressions are analyzed by qPCR. (**C**) BMDMs treated with or without compactin are stimulated by IL-4 (20 ng/ml, 6 hr) and gene expressions are analyzed by qPCR. (**D**) Chromatin-immunoprecipitation (ChIP) analysis of H3K27me3 in BMDMs with or without compactin treatment. The inactive gene *desert* is used as input control. Data are representative of at least 2 independent experiments with 3 samples per group and data are presented as mean ± standrad deviation (SD). Statistical analysis was performed using unpaired, two-tailed Student's *t*-test. Asterisks (*) and (**) indicate a significant difference with $p < 0.05$ and $p < 0.001$. A hashtag (#) indicates not significant.

mammalian cells is in equilibrium with overall cellular cholesterol and, as such, fluctuates with cellular cholesterol through a dynamic steady state (*Steck and Lange, 2010*). Our study here for the first time suggests the mitochondrial membrane as a locus sensing the cellular cholesterol level, which in turn contributes to the regulation of metabolic processes and gene expression. This is somewhat analogous to the regulation of sterol regulatory element-binding proteins (SREBP) pathways by the cellular cholesterol level through the endoplasmic reticulum membrane (*Brown and Goldstein, 1999*). We speculate that the mitochondria likely function far beyond the traditionally called powerhouse that produces ATP (*Shen et al., 2022*).

Macrophages are a major component in both the innate and adaptive immune systems. The anti-inflammatory effect of statins is essentially due to their primary pharmacological action discovered in 1970s, i.e., the inhibition of HMG-CoA reductase (*Endo et al., 1976*), decreasing the level of circulating LDL and inhibiting de novo synthesis of cholesterol in the macrophages themselves. Macrophages with less cholesterol are anti-inflammatory, thereby contributing to the anti-inflammatory action of statins.

## Materials and methods
### Reagents and chemicals
Cell culture Dulbecco's modified Eagle medium (DMEM) was purchased from Gibco (Thermo Fisher Scientific, Watham, MA). Antibiotics (penicillin and streptomycin) and fatty acid-free bovine serum albumin (BSA) were purchased from Sigma-Aldrich (St Louis, MO). Fetal bovine serum (FBS) (Optima) was purchased from Atlanta Biologicals (R&D systems; Minneapolis, MN). As for the chemicals, the following inhibitors: MG-132 (proteasome inhibitor), BAY11-7082 (IKK inhibitor), and BAM15 (another mitochondrial uncoupler) were purchased from TOCRIS chemicals (part of R&D systems). MCD, simvastatin, mevalonate, oligomycin, rotenone, antimycin A, carbonyl cyanide 3-chlorophenylhydrazone (CCCP), 4-(2-hydroxyethyl)-1-piperazineethanesulfonic acid (HEPES), and phosphate-buffered saline (PBS) were purchased from Sigma-Aldrich. For the IMVs study, magnesium chloride ($MgCl_2$), 2-(*N*-morpholino) ethanesulfonic (MES), sucrose, hexane, BSA, sulfuric acid ($H_2SO_4$), hydrochloric acid 37% (HCl), sodium chloride (NaCl), potassium chloride (KCl), dibasic potassium phosphate ($K_2HPO_4$), sodium hydroxide (NaOH), Hydrogen peroxide ($H_2O_2$), trichloroacetic acid (TCA), adenosine 5′-triphosphate disodium salt hydrate ($Na_2ATP$), adenosine 5′-diphosphate sodium salt ($Na_2ADP$), and cholesterol were also supplied by Sigma-Aldrich. Ammonium molybdate (VI) tetrahydrate and L-ascorbic acid were purchased from Acros Organics (part of Sigma-Aldrich). Ultrapure water was produced from a Milli-Q unit (Millipore, conductivity lower than 18 MΩ cm). Rabbit anti-mouse JMJD3 primary antibody was lab-generated as described in *Benyoucef et al., 2016*. The following antibodies were acquired from several vendors: mouse monoclonal anti-β-actin antibody (A1978) from Sigma-Aldrich, rabbit anti-mouse pCREB (87G3), rabbit anti-mouse CREB (86B10), and rabbit anti-mouse Tri-Methyl-Histone H3 (Lys27) antibodies were from Cell Signaling Technology (Danvers, MA). The secondary antibody Horseradish peroxidase (HRP)-conjugated anti-rabbit antibody was from Cayman chemicals (Ann Arbor, MI). The enhanced chemiluminescence (ECL) solutions for the western blotting system were from GE Healthcare (Chicago, IL). The protease and phosphatase inhibitor cocktails were purchased from Sigma-Aldrich.

### RAW264.7
RAW 264.7 TIB-71 cells were directly ordered from American Type Collection Culture (ATCC) and authenticated using STR. Cells were tested for 'Mycoplasma contamination' by ATCC Cell Authentication Service in October 2023 and result was negative. Cells were maintained in 100-mm diameter tissue culture-treated polystyrene dishes (Fisher Scientific, Hampton, NH) at 37°C in a humidified atmosphere

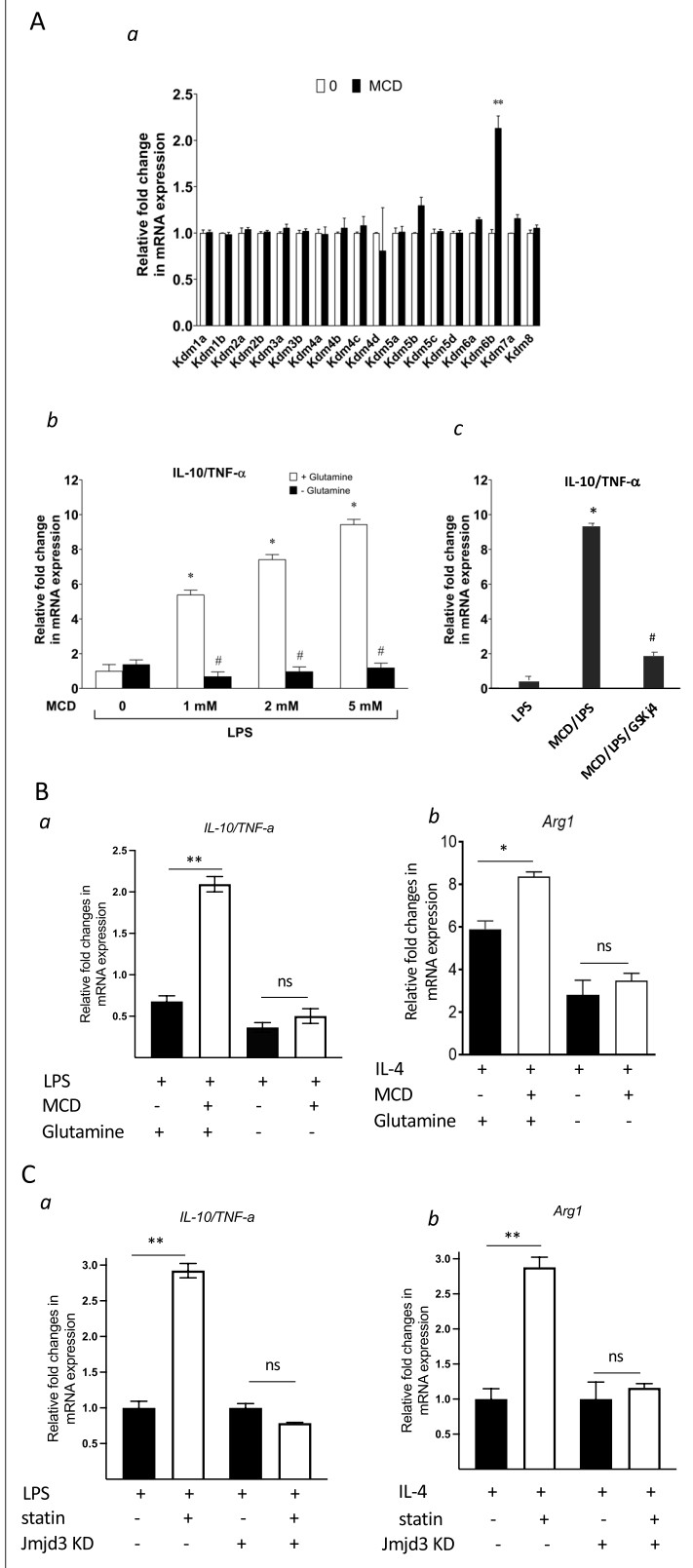

**Figure 8.** Cholesterol reduction suppresses proinflammatory phenotypes and enhances the expression of anti-inflammatory factors, depending on *Jmjd3* and its enzymatic activity. (**A**) RAW macrophages were treated with methyl-β-cyclodextrin (MCD) (5 mM, 1 hr). (**a**) The expression of JmjC demethylase family with or without MCD treatment. (**b**) After lipopolysaccharide (LPS) stimulation (100 ng/ml, 3 hr), in the presence of glutamine

*Figure 8 continued on next page*

*Figure 8 continued*

or without, the expression of *Il10* and *Tnfa* was analyzed by qPCR to generate the ratio of *Il10*/*Tnfa*. (**c**) Effect of GSJK4, a JMJD3 inhibitor, on *Il10*/*Tnfa*. (**B**) (**a**) Bone marrow-derived macrophages (BMDMs) treated with or without MCD are stimulated by LPS (50 ng/ml, 3 hr) in the presence of glutamine or without. Expression of *Il10* and *Tnfa* was analyzed by qPCR to generate the ratio of ratio of *Il10*/*Tnfa*. (**b**) BMDMs treated with or without MCD are stimulated by IL-4 (20 ng/ml, 6 hr) and expression of *Arg1* is analyzed by qPCR. (**C**) wt and *Jmjd3* KD RAW macrophages were treated with compactin (10 + 200 µM mevalonate; 2 days) and then stimulated with LPS (100 ng/ml, 3 hr) or IL-4 (20 ng/ml, 6 hr). Expression of *Il10* and *Tnfa* (**a**) or *Arg1* (**b**) was analyzed by qPCR. Data are representative of 3 independent experiments with 3 samples per group and data are presented as mean ± standard deviation (SD). Statistical analysis was performed using unpaired, two-tailed Student's *t*-test. Asterisks (*) and (**) indicate a significant difference with $p < 0.05$ and $p < 0.001$. A hashtag (#) indicates a significant difference between MCD without or with glutamine with $p < 0.05$.

The online version of this article includes the following figure supplement(s) for figure 8:

**Figure supplement 1.** Expression of the Jmjd3.

**Figure supplement 2.** Expression of JMJD3 in wt and Jmjd3 KD macrophages (mean ± standard deviation [SD]), $p < 0.001$.

of 95% air and 5% $CO_2$. The cells were cultured in DMEM-based growth medium containing glutamine and without pyruvate, supplemented with 10% (vol/vol) heat-inactivated FBS and 1× penicillin/streptomycin. For experiments (and routine subculture), cells were collected in growth media after 25-min incubation with Accutase (Sigma-Aldrich, St Louis, MO) at 37°C in a humidified atmosphere of 95% air and 5% $CO_2$. In preparation for experiments, cells were seeded in 6-well tissue culture-treated polystyrene plates (Fisher Scientific, Hampton, NH) at 30,000 cells/$cm^2$ for 2 days until cells have recovered in time for treatments. Unless otherwise indicated, cell treatments were prepared in DMEM supplemented with 0.25% (wt/vol) FA-free BSA that was sterilized by filtration through 0.2 µm pore size cellulose acetate syringe filters. For treatments, cells were washed with pre-warmed sterile PBS and were treated with media containing MCD (5 mM), cell culture-grade water (negative control). Then, the cells were incubated for 1 hr under cell culture conditions. RAW 264.7 macrophages were pre-incubated for 5 min with CCCP (50 µM), 30 min with BAM15 (200 µM), or 1 hr with NF-κB inhibitors (MG-132 [5 µM] or BAY11-7082 [10 µM]).

## Bone marrow-derived macrophages

BMDMs were differentiated from bone marrow cells isolated from the femora, and tibiae of 4- to 16-week-old wild type C57BL/6J mice. Euthanasia was performed by $CO_2$ gas asphyxiation followed by cervical dislocation. Euthanized mice were soaked with 70% (vol/vol) ethanol immediately prior to dissection. After careful isolation of the bones, the ends of the bones were cut with sterile scissors and centrifuged at 10,000 rpm for 15 s at room temperature in a microcentrifuge (MCT) tube where the bone marrow is collected. The bone marrow is then filtered using sterile, combined and centrifuged at 480 × *g* for 10 min at room temperature. The cell pellet is resuspended briefly (<2 min) in red blood cell lysis buffer. The differentiation medium (DMEM + 20% L-929-conditioned media + 10% FBS + 1% penicillin–streptomycin) is then added and cells are centrifuged again at 480 × *g* for 10 min at room temperature. The cell suspension is filtered once more through a 70-µm strainer into another 50 ml centrifuge tube to rinse and collect all the cells. 100-mm diameter suspension dishes (Greiner Bio-One, Monroe, NC) are seeded with 10 ml per dish at 0.6–0.8 × $10^6$ cells/ml (~10,000–15,000 cells/$cm^2$). The dishes were then incubated for 6 days at 37°C in a humidified atmosphere of 95% air and 5% $CO_2$. Differentiation media (10 ml) are added on day 3 or 4.

At the end of day 7, cells are detached with trypsin, counted and reseeded into 6-well plates at 0.5 million cells/well (~50,000 cells/$cm^2$) using 2 ml/well of DMEM supplemented with 10% FBS and 1% P/S. Plates are left overnight at 37°C in a humidified atmosphere of 95% air and 5% $CO_2$ to ensure cell adherence. BMDMs are now ready to be used for assays. Similar to RAW264.7 cells, cell treatments were prepared in DMEM supplemented with 0.25% (wt/vol) FA-free BSA. For treatments, cells were washed with pre-warmed sterile PBS and were treated with media containing MCD (5 mM), cell culture-grade water (negative control). Then, the cells were incubated for 1 hr under cell culture conditions.

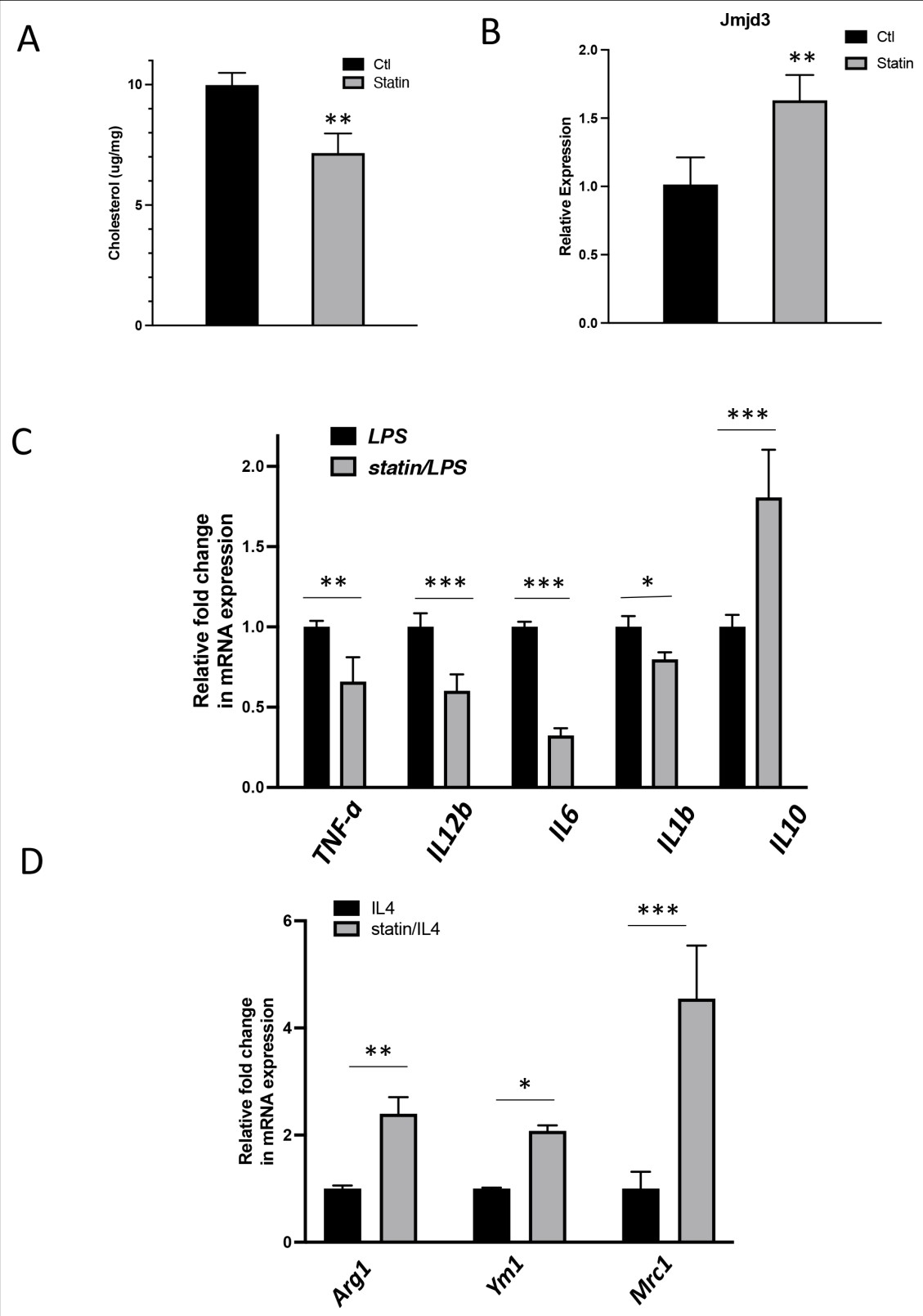

**Figure 9.** In vivo statin feeding in mice reduces cholesterol and upregulates *Jmjd3* in the peritoneal macrophages, which conveys anti-inflammatory phenotype. (**A**) Total cholesterol contents (free and esterified) in freshly isolated mouse peritoneal macrophages from simvastatin-fed (100 μg/kg/day, 14 days) and control mice. (**B**) *Jmjd3* gene expression in peritoneal macrophages from simvastatin-fed (statin) and control mice. (**C**) Freshly isolated mouse peritoneal macrophages from simvastatin-fed and control mice are stimulated by lipopolysaccharide (LPS) (100 ng/ml, 6 hr). Gene expressions

*Figure 9 continued on next page*

*Figure 9 continued*

are analyzed by qPCR. (**D**) Freshly isolated mouse peritoneal macrophages from simvastatin-fed and control mice are stimulated by IL-4 (20 ng/ml, 6 hr) and gene expressions are analyzed by qPCR. Data are representative of 3 independent experiments with 3–4 samples per group and presented as mean ± standard deviation (SD). Statistical analysis was performed using unpaired, two-tailed Student's *t*-test. Asterisks (*), (**), and (***) indicate a significant difference with $p < 0.05$, $p < 0.005$, and $p < 0.001$, respectively.

With either BMDMs or RAW264.7, for LPS or Il-4 stimulation, cells were washed with sterile PBS post MCD or statins treatment and before adding LPS (100 ng/ml) or IL-4 (20 ng/ml) for an additional 3h incubation.

## JMJD3 shRNA preparation

For the JMJD3-shRNA construct, the ShKDM6B-265-Up CCGG CCTCTGTTCTTGAGGGACAAA CTCGAG TTTGTCCCTCAAGAACAGAGG TTTTTG and ShKDM6B-265-Down AATTCAAAAA CCTC TGTTCTTGAGGGACAAA CTCGAG TTTGTCCCTCAAGAACAGAGG sequences were used to generate lentivirus harboring shRNA or empty Neo (control). The shRNAs were cloned in lab into pLKO.2 Neo plasmid using EcoR1 and AgeI restriction enzymes. The JMJD3 shRNA lentiviral titer was prepared using a polyethylenimine (PEI)-based transfection. Briefly, on day 1 HEK 293T cells were seeded at $8 \times 10^6$ cells in 15 cm dishes. On day 2 (transfection day), before the start of transfection, media was removed, cells were washed with media/PBS, then fresh media (media containing 5% FBS) is added. 30 μg total DNA was used (17.5 μg of shRNA) to transfect the cells seeded. OptiMEM media was used to prepare the transfection mix where 1 ml DNA is mixed to 1 ml PEI reagent, and incubated for another 20–30 min. The mix is added gently on to the cells, drop by drop. After overnight incubation, the media is replaced and fresh 5% FBS-containing media is added. The next day, i.e., 46-hr post-transfection, the media is collected. This media will contain virus particles. Fresh media is added and collected after 24 hr, i.e. 70-hr post-transfection. The media collected is pooled to process by ultracentrifugation and collect a concentrated titer of virus particles.

## Jmjd3 knockdown in RAW 264.7 macrophages

RAW 264.7 macrophages were transfected with JMJD3 shRNA lentiviral particles for 18–20 hr and then incubated for 48 hr in growth media (+/−statins). Cells were then stimulated with LPS or IL-4 before DNA extraction and qPCR analysis.

## In vivo statin experiment

C57BL/6J mice were fed the chow diet (WQJX Bio-technology) to which simvastatin (100 mg/kg/day; Merck & Co Inc) added for 14 days. Control animals were fed the chow diet. Peritoneal macrophages were harvested from the mice 4 days after 1.5 ml of thioglycolate broth (Sigma) was injected. Cells were washed with PBS, seeded at a density of 1,000,000 per well into 24-well dishes, incubated for 12 hr in DMEM containing 10% lipoprotein serum deficient (LPSD) and 1 μM simvastatin (to maintain in vivo cholesterol levels). Cells were then stimulated with 100 ng/ml LPS (MedChemExpress) or 20 ng/ml IL-4 (Aladdin), and RNA was isolated for qPCR. Cholesterol contents were analyzed with cholesterol quantification assay kit (Sigma).

## RNA purification and cDNA synthesis

After treatment, the cells were lysed and collected in TRIzol (Thermo Fisher Scientific, Watham, MA), and frozen at −80°C. Total RNA was extracted by phenol–chloroform extraction, followed by ethanol precipitation. RNA was purified through columns supplied in the Molecular Biology kit (BioBasic Inc, Markham, ON), according to the manufacturer's instructions. The RNA concentrations and purity were determined using a NanoDrop One (Thermo Fisher Scientific, Watham, MA) spectrophotometer. cDNA synthesis was performed on the Bio-Rad T100 PCR Gradient Thermal Cycler using the QuantiTect Reverse Transcription kit (QIAGEN, Germantown, MD), following the manufacturer's instructions.

## Reverse transcriptase quantitative PCR

Gene expression was analyzed by real-time reverse transcriptase quantitative PCR according to the Fast SYBR Green protocol with the AriaMx real-time PCR detection system (Agilent technologies, Santa Clara, CA). Primers were ordered from Invitrogen (Thermo Fisher Scientific, Watham, MA) and

**Table 2.** Mouse primers used for real-time RT-qPCR.

| Primer | | Sequence |
|--------|--------|-----------|
| Hprt1 | Forward | TGTTGTTGGATATGCCCTTG |
| | Reverse | TTGCGCTCATCTTAGGCTTT |
| Il1b | Forward | AGTTGACGGACCCCAAAAGAT |
| | Reverse | GTTGATGTGCTGCTGGGAGA |
| Jmjd3 | Forward | CCAGGCCACCAAGAGAATAA |
| | Reverse | CGCTGATGGTCTCCCAATAG |
| Tnfa | Forward | CCGTAGGGCGATTACAGTCA |
| | Reverse | CCTGGCCTCTCTACCTTGTTG |
| Il10 | Forward | TGGCCCAGAAATCAAGGAGC |
| | Reverse | CAGCAGACTCAATACACACT |
| Il6 | Forward | TAGTCCTTCCTACCCCAATTTCC |
| | Reverse | TTGGTCCTTAGCCACTCCTTC |
| Il12b | Forward | GGAAGCACGGCAGCAGAATA |
| | Reverse | AACTTGAGGGAGAAGTAGGAATGG |
| Arg1 | Forward | CTCCAAGCCAAAGTCCTTAGAG |
| | Reverse | AGGAGCTGTCATTAGGGACATC |
| Ym1 | Forward | AGAAGGGAGTTTCAAACCTGGT |
| | Reverse | GTCTTGCTCATGTGTGTAAGTGA |
| Mrc1 | Forward | CTCTGTTCAGCTATTGGACGC |
| | Reverse | CGGAATTTCTGGGATTCAGCTTC |

are listed in *Table 2*. Each condition was prepared in triplicates and each sample was loaded as technical triplicates for each gene (target or reference) analyzed. The mRNA levels of mouse HPRT1 or GAPDH were used as internal controls (reference gene) as indicated.

## Immunoblotting

For western blot analysis, MCD-treated RAW 264.7 cells were washed in ice-cold PBS, lysed in radio-immune precipitation assay buffer (150 mM NaCl, 1% Nonidet P-40, 1% sodium deoxycholate, and 25 mM Tris (pH 7.6)) supplemented with a cocktail of protease and phosphatase inhibitors. Total cell lysis was achieved through sonication for 20 s at 20% amplitude and samples were stored at −80°C. Histone extraction was performed following a specific protocol from Abcam. Protein concentration was determined using the Bradford assay (Bio-Rad, 5000006). Sodium dodecyl sulfate (SDS) buffer (313 mM Tris (pH 6.8), 10% SDS, 0.05% bromophenol blue, 50% glycerol, and 0.1 M dithiothreitol [DTT]) was added, and the samples were boiled at 100°C for 5 min. Proteins were separated by SDS–polyacrylamide gel electrophoresis on 10% acrylamide gels. After separation, proteins were transferred onto polyvinylidene fluoride (PVDF) membranes and were blocked in 5% milk powder (in PBS, 1% Triton X-100) for 1 hr. Membranes were incubated with the primary antibodies in the following conditions: overnight incubation with the anti-Jmjd3 antibody (1:400) at 4°C, and 1 hr with the anti-H3K27Me3 antibody at room temperature. After washes, blots were further incubated with an HRP-conjugated anti-rabbit antibody (1:10 000) for 1 hr. Blots were imaged using ECL-based film detection system.

## MCD/cholesterol (10:1 mol/mol) preparation

Cholesterol (0.3 mmol) of is dissolved in 1 ml chloroform solution and dried under nitrogen in a glass culture tube. 10 ml of a MCD solution (300 mM) prepared in BSA/BSS buffer solution (20 mM HEPES, pH 7.4, 135 mM NaCl, 5 mM KCl, 1.8 mM CaCl$_2$, 1 mM MgCl$_2$, 5.6 mM glucose, and 1 mg/ml BSA) was

added to the tube, and the resulting suspension was vortexed, and bath sonicated at 37°C until the suspension clarified. The suspension was then incubated in a rocking water bath overnight at 37°C to maximize formation of soluble complexes. This is the stock solution that can be diluted into desired concentrations.

## Macrophage cholesterol depletion and repletion

On the day of experiment, BMDMs are incubated with 5 mM MCD for 1 hr. The cells are rinsed in 0.25% BSA in DMEM and left to rest for 10 min before switching the media to media containing 1 mM MCD/cholesterol complex for 1 hr. RNAs are collected at the end of incubation for *Jmjd3* expression. Controls cells will be cells that did not receive MCD nor MCD/cholesterol complex. Triplicate wells are prepared and Jmjd3 expression was determined by RT-PCR.

## NF-κB activation assay

NF-κB activation was determined by performing the QUANTI-Blue assay using the RAW-Blue cells (InvivoGen). Through exposure of the RAW-Blue cells to various substances, NF-κB/AP-1 activation is induced, making secreted embryonic alkaline phosphatase (SEAP) into the cell supernatant. In brief, RAW-Blue cells were seeded in 6-well plates ($1.5 \times 10^5$ cells/ml) and were allowed to recover for 2 days in growth media (similar to RAW264.7 cells) before they were stimulated with LPS (100 ng/ml) or MCD (5 mM) for 3 hr. Then, the cells and medium for each sample were collected and sonicated on ice for 15 s at 20% amplitude. The supernatant was separated from cell debris by centrifugation at 4°C for 10 min at 10,000 rpm. For detection, the cell supernatant and an SEAP detection reagent (QUANTI-Blue) were mixed and the absorbance was measured at 650 nm as described in the assay instructions. Each condition was performed in triplicate samples and each sample was measured in technical triplicates.

## Mitochondrial oxygen consumption

Cellular OCRs were measured using an extracellular flux analyzer (Seahorse XF96e; Agilent Technologies, Santa Clara, CA). Briefly, the cartridge sensors were incubated overnight at 37°C in a hydration/calibration solution (XF Calibrant; Agilent Technologies). Specially designed polystyrene tissue culture-treated 96-well microplates with a clear flat bottom (Seahorse XF96 V3 PS Cell Culture Microplates; Agilent Technologies) were then seeded with 80 µl of cell suspension ($4.4 \times 10^5$ cells/ml) per well and incubated 2–3 hr under cell culture conditions to allow cell attachment. For treatment with MCD, cells were washed with pre-warmed sterile PBS and the culture supernatants were replaced with medium containing MCD (5 mM). Then, the cells were incubated for 1 hr under cell culture conditions. For treatment with statins, cells were washed with pre-warmed sterile PBS and the culture supernatants were replaced with medium supplemented with lipoprotein-deficient serum, containing statins (7 mM lovastatin + 200 mM mevalonate). Then, the cells were incubated for 2 days under cell culture conditions. At the end of either treatment, the cells are washed and incubated 45 min at 37°C in extracellular flux analysis medium (sodium bicarbonate-, glucose-, phenol red-, pyruvate-, and glutamine-free modified DMEM [Sigma-Aldrich, catalog no. D5030] freshly supplemented with cell culture-grade D-glucose (4.5 g/L), cell culture-grade L-glutamine [4 mM; Wisent], and HEPES [4.5 mM; Sigma-Aldrich], pH adjusted to 7.35–7.40 at room temperature). OCR was measured to assess resting respiration, followed by ATP production-dependent respiration, maximal respiration, and non-mitochondrial oxygen consumption, after sequential injections of oligomycin (an ATP synthase inhibitor) at 1 µM (final concentration), CCCP (an ionophore acting as a proton uncoupler) at 2 µM, and rotenone together with antimycin A (complex I and III inhibitors, respectively), each at 0.5 µM. Mitochondrial ATP production-dependent respiration was calculated by subtracting the lowest OCR after oligomycin injection from resting OCR. For each parameter, OCR measurements were performed at least three times at 6-min intervals. Each condition was performed in 7–8 replicate samples. All OCR measurements were corrected for the OCR of cell-free wells containing only medium. Upon completion of the OCR measurements, the cells were washed once with PBS and lysed in 1 M NaOH (40 µl/well). The lysates were kept at 4°C for up to 24 hr, and protein determination was performed using the Bradford colorimetric assay with BSA as the standard protein (Thermo Scientific). Absorbance was measured at a wavelength of 595 nm using a hybrid microplate reader (Biotek).

## 27-HC quantification

Cell pellets were collected from RAW264.7 macrophages treated with 1, 3, and 5 mM of MCD for 1 hr. Samples were sent to the University of Texas (UT) Southwestern Medical center for analysis by ultraperformance liquid chromatography/electrospray ionization/tandem mass spectrometry allowing chromatographic resolution of the hydroxycholesterol species. 27-HC levels were calculated by normalizing the 27-HC amounts to the protein content in the whole-cell pellet.

## Purification and formation of *E. coli* IMVs

*E. coli* inner membranes were purified from *E. coli* MG1655 strain by sucrose gradient ultracentrifugation (30 min at 80,000 rpm, Optima MAX-XP) as previously described (*Gutiérrez-Sanz et al., 2016*). The *E. coli* inner membranes were then resuspended in either the synthesis buffer (MES 100 mM, pH 6 and NaCl 25 mM) or the hydrolysis buffer (HEPES 25 mM, pH 8) and ultrasonicated for 15 min (5 s on–off cycles at 30% power, VCX500 Ultrasonic Processors). The resulting homogeneous solution contained the IMVs.

## Loading and depletion of cholesterol in IMVs

Prior to any treatment, the protein concentration of IMVs was determined by the BCA protein assay and IMVs were diluted to a final protein concentration of 0.5 mg/ml. IMVs were then loaded with cholesterol using a modified protocol based on the MCD/cholesterol complex (*Mahammad and Parmryd, 2015a*). To load IMVs, 25 mM MCD were mixed with 2.5 mg/ml of cholesterol on HEPES (25 mM, pH7.8) following the protocol described in *Bacia et al., 2004*. After incubation with MCD–cholesterol complexes, IMVs were ultracentrifuged (80,000 rpm for 30 min) and the supernatant removed. The IMVs were then resuspended in HEPES 25 mM (pH 7.8). After loading, the lipid, cholesterol, and protein concentration of IMVs were determined by the phosphate determination assay (*Chen et al., 1956*), Amplex Red cholesterol assay kit (*Amundson and Zhou, 1999*) and BCA protein assay (*Smith et al., 1985*), respectively. To achieve vesicles with different cholesterol content, cholesterol-loaded IMVs (1 mg/ml protein concentration) were treated with increasing MCD concentrations (from 0 to 7.0 mM) for 30 min at 37°C[3,] (*Mahammad and Parmryd, 2015a*; *Pucadyil and Chattopadhyay, 2007*). Cholesterol-depleted vesicles were ultracentrifuged (80,000 rpm for 30 min) and the pellet was resuspended in HEPES (25 mM, pH = 7.8). Lipid, protein, and cholesterol concentration of depleted samples were quantified again after the MCD treatment.

## Hydrolysis and synthesis of ATP on cholesterol-doped IMVs

ATP hydrolysis was performed by adding a total concentration of 2 mM ATP to 200 mM IMVs (lipid concentration) and incubated for 30 min. The concentration of phosphates from ATP hydrolysis was measured using the malaquite green assay (*Lanzetta et al., 1979*). ATP synthesis was triggered by promoting a ΔpH across the IMV membranes by mixing the samples (IMVs resuspended in MES 100 mM, pH 6) with an external buffer with higher pH and in the presence of inorganic phosphate, ADP, and magnesium ion (HEPES 100 mM pH 8, 5 mM $P_i$, 2.5 mM ADP, 5 mM $MgCl_2$, 25 mM NaCl) at a volume ratio of 1:10 and incubated for 2–5 min. The reaction was stopped by adding 20% TCA at a ratio volume of 10:1, and then the samples were equilibrated to neutral pH (*Kaim and Dimroth, 1994*). ATP concentration after synthesis was measured using ATP detection assay kit (Molecular Probes) with a luminometer GloMax-Multi Detection. Amplex Red Cholesterol Assay Kit and Pierce BCA Protein assay kits were supplied by Thermo Fisher. Luminescent ATP Detection Assay Kit (based on firefly's luciferase/luciferin) was purchased from Molecular Probes.

## RNA-seq and data analysis

RAW 264.7 macrophages in duplicates were either left untreated or treated with 5 mM MCD for 1 hr. Total RNA was purified as described above, and the RNA concentrations and purity were determined using a NanoDrop One (Thermo Fisher Scientific, Watham, MA) spectrophotometer. Library preparation and 150 bp paired-end RNA-seq were performed using standard Illumina procedures for the NextSeq 500 platform. Reads libraries produced from RNA sequencing for each replicate and condition were aligned against mouse genome reference provided by the GENCODE project – the mouse genome reference release M25 (*GENCODE Consortium, 2021*). Transcript quantification count tables were generated per bait screening using the Salmon algorithm ver1.7.0 (*Patro et al.,*

*2017*). The following comparisons: MCD1 vs. Control, Statin vs. Control, and Statin-LPS vs. LPS were performed to identify DEGs using DESeq2 ver1.40.2 (*Love et al., 2014*).

## GO enrichment and Kyoto encyclopedia of genes and genomes (KEGG) pathway analyses

GO functional enrichment and KEGG pathway analyses for the DEGs were performed using cluster-Profiler ver4.8.2 (*Wu et al., 2021*) to identify significantly enriched biological processes associated with the set of genes upregulated in the experimental condition greater than $\log_2$ fold change of 1 and with the set of genes downregulated with $\log_2$ fold change of less than −1. A p value cutoff of 0.05 was defined for the analyses. The Benjamini–Hochberg method was applied to adjust the p values for multiple testing. This method controls the false discovery rate during the adjustment process.

## ATAC-seq

RAW 264.7 macrophages in duplicates were either left untreated or treated with 5 mM MCD for 1 hr. Cells were detached using 0.5% trypsin and resuspended in chilled 1× PBS containing 1 mM ethylene-diaminetetraacetic acid (EDTA). Visible cell numbers per sample were obtained by staining the cells with trypan blue and counting them using hemocytometer. 50,000 cells were used for performing ATAC-seq. Cells were washed in 100 µl of ice-cold 1× PBS and centrifuged at 500 × *g* for 5 min. Nuclei are prepared by lysing the cells in ice-cold lysis buffer containing 10 mM Tris–Cl, pH 7.4, 10 mM NaCl, 3 mM MgCl$_2$, and 0.1% IGEPAL CA-630. Nuclei were pelleted by spinning the samples at 500 × *g* for 10 min in fixed-angle cold centrifuge and proceeded for tagmentation reaction. A 25-µl tagmentation reaction was setup by resuspending the nuclei in 12.5 µl of 2× Tagment DNA buffer and 5 µl of TDE1 transposase. Samples were incubated at 37°C for 30 min with gentle intermittent mixing. Following transposition, the sample volume was made up to 50 µl using resuspension buffer and was processed for DNA preparation. For tagmented DNA clean-up, 180 µl of Zymo DNA-binding buffer was added and mixed thoroughly before loading on to Zymo-spin concentrator-5 columns. For transposase-free DNA, the samples were eluted in 16.5 µl of elution buffer. Purified tagmented DNA was amplified with KAPA HIFI polymerase (12 PCR cycles) and Unique Dual Primers from Illumina (Cat number 20332088). Size selection (L:1.1; R:0.6) was performed with KAPA Pure beads and size distribution of the final libraries was assessed on bioanalyzer (Agilent). Libraries were quantified by qPCR and loaded equi-molarly on a S4 Novaseq flowcell. Each library was sequenced with a coverage of 50 M paired-end reads (PE100).

## ATAC-seq analysis

Sequencing reads for chromatin accessibility (ATAC) were aligned to *Mus musculus* genome assembly GRCm38 (mm10) using Bowtie2 (*Langmead and Salzberg, 2012*) with default parameters. The resulting BAM files were filtered to remove duplicate reads using Picard Tools (https://broadinstitute.github.io/picard/). Peaks were called using MACS2 ver2.0 (*Zhang et al., 2008*) with the parameter '-nomodel'. The generated narrow peaks files were used for downstream analysis. Diffbind ver3.10 (*Stark and Brown, 2011*; *Ross-Innes et al., 2012*) was used to identify differentially accessible regions (peaks) called by MACS2. Comparison between the two replicates of the two conditions MCD1 vs. Control was conducted. DESeq2 ver1.40.2 (*Love et al., 2014*) and EdgeR ver3.42.4 (*Robinson et al., 2010*) were used within Diffbind to identify regions of differential accessible between MCD1 treated and control (p value <0.05 and RD <0.05). ChipSeeker ver1.3 (*Wang et al., 2022*; *Yu et al., 2015*) was used to annotate the genomic features of the differentially accessible peaks identified by Diffbind, where the maximum range of promoter to transcription start site was set to 3 kb. The peaks were assigned to the nearest genes based on distance of the peak region to the transcription start site. This allowed the annotation of ATAC-seq peaks with genes. The heatmap was generated using Diffbind::dba.plotProfile in order to compute peakset profiles for MCD1 and control conditions of loss or gain of genomic accessibility.

## Gene set enrichment analysis

Functional gene set enrichment analysis (FGSEA) was conducted for all RNA-seq comparisons and ATAC-seq comparison. FGSEA was conducted using fgsea ver1.26 (*Korotkevich et al., 2019*). MSigDB gene sets utilized in fgsea were Hallmark gene sets ver7.1, C2 BioCarta pathways ver7.1, C2 KEGG

pathways ver7.1, C2 Reactome pathways ver7.1, and C3 Transcription factor targets ver7.1. The gene list provided to fgsea was based on the DEGs, including both up- and downregulated genes ranked in descending order or log$_2$ fold change. The size of the gene sets considered for the enrichment analysis was set to a minimum of 15 and maximum of 500. Only Hallmark pathways with adjusted p value of less than 0.01 was plotted. Enrichment plots for the Hallmark pathway with the highest normalized enrichment score are illustrated.

## Statistical analysis

Statistical analyses between data groups were performed with PRISM software (GraphPad). Data for real-time RT-qPCR and Seahorse experiments are presented as the mean ± standard deviation as indicated. The statistical significance of differences between groups was analyzed by Student's *t*-test. Differences were considered significant at a *p* value <0.05.

## Acknowledgements

This work was supported by a grant (grant-in-aid) from Heart and Stroke Foundation of Canada (HSFC), G-19-0026359, and a grant from Canadian Institute of Health Research (CIHR), PJT-180504. This work was also supported in part by the TECNOLOGÍAS 2018 program funded by the Regional Government of Madrid (Grant S2018/BAA-4403 SINOXPHOS-CM, to ILM) and by a grant from NIH (HL020948, to JGM). We thank Dr. T Lagace for reagents, and Dr. M Harper for access to use the XFe96 Seahorse Extracellular Flux Analyzer. We also thank the Ottawa Hospital Research Institute's Biotherapeutic Manufacturing Centre-Virus Manufacturing Facility for making shRNA lentivirus. We knowledge the technical assistance of H Bandukwala, Yuefeng Li, and E S Qamsar.

## Additional information

### Funding

| Funder | Grant reference number | Author |
|---|---|---|
| Heart and Stroke Foundation of Canada | G-19-0026359 | Xiaohui Zha |
| Canadian Institutes of Health Research | PJT-180504 | Jeffrey F Dilworth |
| National Institutes of Health | HL020948 | Jeffrey McDonald |
| Regional Government of Madrid | S2018/BAA-4403SINOXPHOS-CM | Iván López-Montero |

The funders had no role in study design, data collection, and interpretation, or the decision to submit the work for publication.

### Author contributions

Zeina Salloum, Xiaohui Zha, Conceptualization, Data curation, Formal analysis, Supervision, Funding acquisition, Validation, Investigation, Visualization, Methodology, Writing – original draft, Project administration, Writing – review and editing; Kristin Dauner, Conceptualization, Data curation, Formal analysis, Validation, Investigation, Visualization, Methodology, Writing – original draft, Project administration, Writing – review and editing; Yun-feng Li, Conceptualization, Data curation, Formal analysis, Investigation, Methodology; Neha Verma, Data curation, Investigation; David Valdivieso-González, Víctor Almendro-Vedia, John D Zhang, Kiran Nakka, Mei Xi Chen, Data curation, Formal analysis, Investigation; Jeffrey McDonald, Chase D Corley, Data curation, Formal analysis; Alexander Sorisky, Conceptualization, Data curation, Formal analysis, Writing – original draft, Writing – review and editing; Bao-Liang Song, Conceptualization, Data curation, Supervision, Writing – original draft, Writing – review and editing; Iván López-Montero, Jie Luo, Data curation, Formal analysis, Supervision, Funding acquisition, Investigation, Methodology, Writing – original draft, Writing – review and editing; Jeffrey F Dilworth, Conceptualization, Data curation,

Supervision, Funding acquisition, Investigation, Methodology, Writing – original draft, Writing – review and editing

### Author ORCIDs
Zeina Salloum https://orcid.org/0000-0002-5216-8601
Víctor Almendro-Vedia https://orcid.org/0000-0002-7297-1901
Kiran Nakka https://orcid.org/0000-0002-8418-9343
Iván López-Montero https://orcid.org/0000-0001-8131-6063
Xiaohui Zha https://orcid.org/0000-0003-2873-3073

### Decision letter and Author response
Decision letter https://doi.org/10.7554/eLife.85964.sa1
Author response https://doi.org/10.7554/eLife.85964.sa2

## Additional files

### Supplementary files
• MDAR checklist

• Supplementary file 1. RNA-seq data of macrophages treated or not with statins.

• Supplementary file 2. RNA-seq data of macrophages treated or not with MCD.

• Supplementary file 3. ATAC-seq data of macrophages treated or not with MCD.

• Supplementary file 4. Genes showing chromatin opening (ATAC-Seq peak) and increased expression (RNA-Seq) after MCD treatment as well as Statin Treatment.

• Supplementary file 5. RNA-seq of macrophages treated with or not with statins and then stimulated by LPS.

### Data availability
Sequencing data have been deposited in GEO under accession codes: GSE196187, GSE196188, GSE196189.2. All data generated or analyzed during this study are included in the manuscript.

The following datasets were generated:

| Author(s) | Year | Dataset title | Dataset URL | Database and Identifier |
|---|---|---|---|---|
| Salloum Z, Dauner K, Verma N, Zhang JD, Nakka K, Chen MX, Valdivieso-González D, Almendro-Vedia V, McDonald J, Corley CD, Sorisky A, Song BL, Li YF | 2024 | Data for: Statin-mediated reduction in mitochondrial cholesterol primes an anti-inflammatory response in macrophages by upregulating Jmjd3 | https://www.ncbi.nlm.nih.gov/geo/query/acc.cgi?acc=GSE196187 | NCBI Gene Expression Omnibus, GSE196187 |
| Salloum Z, Dauner K, Verma N, Zhang JD, Nakka K, Chen MX, Valdivieso-González D, Almendro-Vedia V, McDonald J, Corley CD, Sorisky A, Song BL, López-Montero I, Luo J, Dilworth JF, Zha X, Li YF | 2024 | Data for: Statin-mediated reduction in mitochondrial cholesterol primes an anti-inflammatory response in macrophages by upregulating Jmjd3 | https://www.ncbi.nlm.nih.gov/geo/query/acc.cgi?acc=GSE196188 | NCBI Gene Expression Omnibus, GSE196188 |
| Salloum Z, Dauner K, Verma N, Zhang JD, Nakka K, Chen MX, Valdivieso-González D, Almendro-Vedia V, McDonald J, Corley CD, Sorisky A, Song BL, López-Montero I, Luo J, Dilworth JF, Zha X, Li YF | 2024 | Data for: Statin-mediated reduction in mitochondrial cholesterol primes an anti-inflammatory response in macrophages by upregulating Jmjd3 | https://www.ncbi.nlm.nih.gov/geo/query/acc.cgi?acc=GSE196189 | NCBI Gene Expression Omnibus, GSE196189 |

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
