## [Editor Report]

The manuscript by Salloum and colleagues examines the role of statin-mediated regulation of mitochondrial cholesterol as a determinant of epigenetic programming via JMJD3 in macrophages. The findings could be valuable for understanding statin-mediated anti-inflammatory functions in macrophages. The evidence for a statin-cholesterol-JMJD3-H3K27 axis as a regulator of macrophage function is solid.

---

## [Decision Letter]

**Decision letter after peer review:**

Thank you for submitting your article "Statin-mediated reduction in mitochondrial cholesterol primes an anti-inflammatory response in macrophages by upregulating JMJD3" for consideration by *eLife*. Your article has been reviewed by 3 peer reviewers, including Jalees Rehman as the Reviewing Editor and Reviewer #1, and the evaluation has been overseen by Satyajit Rath as the Senior Editor. The following individual involved in the review of your submission has agreed to reveal their identity: Julia Sánchez-Ceinos (Reviewer #3).

Essential revisions (for the authors):

1) Need for in-depth analysis and presentation of the RNA-seq and ATAC-seq data

2) Mechanistic studies using more specific treatments such as genetic silencing of ATP synthase and JMJD3 (using siRNA, shRNA, or CRISPR with appropriate controls) and the use of additional NFKB inhibitors to ensure the specificity of the findings

3) Demonstrate in vivo relevance of the observed statin effects in order to justify that this is indeed a mechanism by which statins can affect inflammation

4) Provide details on which statins were used, which concentrations, and treatment durations

*Reviewer #1 (Recommendations for the authors):*

1. in vivo treatment with statins (either humans or animal models) versus control, followed by isolation of macrophages to assess gene expression, mitochondrial function, and mitochondrial membrane composition would be important to support the core conclusions

2. Instead of using CCCP, selective shRNA depletion of ATP synthase would provide more targeted information without potential mitochondrial toxicity and ROS generation that occurs with CCCP.

3. The RNA-seq and ATAC-seq data are presented in a very limited and superficial manner. Instead of showing only the inferred transcription factor binding sites, it would be helpful to first show the most upregulated and downregulated genes and gene expression pathways in the RNA-seq data. Similar in-depth analysis of the ATAC-seq data would be helpful, and further determination of concordance versus discordance between the RNA-seq and ATAC-seq data. Importantly, as H3K27 methylation is a key part of the conclusions, it would be important to perform ChIP-seq for H3K27 methylation instead of selected PCR. This would provide a comprehensive insight into which gene programs are affected. These in-depth bioinformatic analyses should be accompanied by the descriptions in the Methods sections

4. Statin experiments need to indicate which statins were used, which concentrations, and how long cells were treated. This is essential to ensure reproducibility and comparison with other statin studies.

*Reviewer #2 (Recommendations for the authors):*

The authors would benefit from thinking about their experiments and what they do or do not clearly show. It is clear that JMJD3 is being upregulated by statins and MCD however, some of the mechanistic claims that they are making feel a little too preliminary.

For example, MG-132 is a well-characterized proteasome inhibitor. There may be a better NFKB inhibitor to use here for this analysis.

The western blots in Figure 1 could be of slightly better quality to really convincingly show that these cholesterol perturbations are changing protein expression. Furthermore, for the TNF and IL1b results, they would be strengthened by showing these perturbations have an impact at the protein level. The authors also don't really discuss the literature about cholesterol and the inflammasome which may be contributing to the results that they are seeing with IL1b.

There are several figures that don't convey a clear message. For example, the presentation of the omics results could be improved in order to better convey a strong message. While it is clear that the statins impact the ATACseq result, whether this is at all related to altered JMJD3 expression is not clear so figure 6 in its current form is divorced from the narrative that was established in the earlier figures.

All small molecule perturbation experiments would benefit from (1) controls to show that the drugs are not toxic at the concentration used and (2) evidence that these results are specific for the target of interest and not just wholesale altering gene expression across the transcriptome.

[Editors' note: further revisions were suggested prior to acceptance, as described below.]

Thank you for resubmitting your work entitled "Statin-mediated reduction in mitochondrial cholesterol primes an anti-inflammatory response in macrophages by upregulating Jmjd3" for further consideration by *eLife*. Your revised article has been evaluated by Satyajit Rath (Senior Editor) and a Reviewing Editor.

The manuscript has been improved but there are some remaining issues that need to be addressed, as outlined below:

Essential revisions that were requested during the last review have not been addressed. This includes some in vivo data to support the key claims of the work as well as a standard bioinformatic analysis which shows all the differentially expressed genes (top 25 or 50 can be shown in heatmaps, full table can be a supplement). The answers that were provided to the essential revisions do not adequately address the reviewer comments. The interpretations and conclusions of the manuscript do not match the data that are presented.

Please review the comments and essential revisions requested from the 1st review and also take into account the comments provided again below:

*Reviewer 1 (Recommendations for the authors):*

1. There are no in vivo studies in the revision even though demonstrating in vivo relevance of the findings would be essential to bolster the claims of the work. There are several animal models available in which statin efficacy has been demonstrated

See Pecoraro et al., 2014:

https://onlinelibrary.wiley.com/doi/10.1111/eci.12304

There are also classic review articles highlighting statin efficacy in mouse models:

https://www.ahajournals.org/doi/10.1161/atvbaha.107.142570

In addition, non-rodent models such as rabbits are also an option.

2. The bioinformatic analysis does not show the various differentially regulated genes nor is there an in-depth analysis of the epigenetic data that was recommended as essential. In-depth analyses recommended listing all the differentially expressed genes – not just selected pathways via GSEA – for RNA, differentially accessible chromatin regions, etc. GSEA just gives a limited perspective and does not convey a sense of the overall data.

*Reviewer #2 (Recommendations for the authors):*

The authors have addressed many of my concerns; however, I suggest they review their language closely in their manuscript one more time to make sure that they are pursuing conservative interpretations of their data without overclaiming.

*Reviewer #3 (Recommendations for the authors):*

Although the findings are potentially of interest to *eLife* readership, regrettably the revised manuscript and author responses fail to address the comments pointed out by this reviewer.

---

## [Author Response]

RNA-seq data from statin- and MCD-treated macrophages was re-analyzed by unsupervised Gene Set Enrichment Analysis (GSEA) (Figure 1 A & B), which includes all differentially expressed genes, up and down, by cholesterol reduction. The conclusion is identical to the previous analysis, i.e. NF-κB is the top pathway activated by cholesterol reduction. The analysis in last version, which used a different program, is now moved to Supplementary Figure 1.ATAC-seq data was similarly re-analyzed with GSEA (Figure 6 A). Again, NF-κB is the top pathway activated by cholesterol reduction (Figure 6 A, b). Examples of the lineups between ATAC-Seq peaks and RNA-seq peaks have been added (Figure 6 B).RNA-seq data from LPS-stimulated macrophages with or without statins is also re-analyzed. Gene Ontology (GO) analysis of genes showing decreased expression upon statin treatment revealed that statins primarily suppress inflammatory processes (Figure 7 A, b), while genes involved in cellular homeostatic functions were upregulated (Figure 7 A, *c*).2) Mechanistic studies using more specific treatments such as genetic silencing of ATP synthase and JMJD3 (using siRNA, shRNA, or CRISPR with appropriate controls) and the use of additional NFKB inhibitors to ensure the specificity of the findings

Jmjd3

Jmjd3 was knocked down in macrophages by siRNA to < 20% of the control value (Supplementary 7). The impact of the knockdown on the statin effect on LPS- or IL-4 stimulated gene expression was tested. The results are identical to that of the glutamine removal strategy (Figure 8 B, a and b). With Jmjd3 KD, statin could no long promote anti-inflammatory IL-10 expression and fail to raise the ratio of TNF-a/IL-10. At the same time, Arg1 expression, which was elevated by statin, remains the same in statin-treated macrophages (Figure 8 C, a and b).

Mitochondria ATP-synthase

We agree with reviewer that directly manipulating ATP synthase in the mitochondria would significantly reinforce our existing observations and conclusions. However, as we have observed OCR by Seahorse, MCD decreases ATP synthase activity by 20% in BMDMs (Figure 3). The quick recovery of activity by cholesterol repletion (Figure 1 D, c) was seen in the case of this degree of inhibition. This reversible perturbance is not likely to occur by silencing ATP synthase – genetic silencing ATP synthase was shown to cause severe mitochondria structure damages (Quintana-Cabrera R. et al. The cristae modulator Optic atrophy 1 requires mitochondrial ATP synthase oligomers to safeguard mitochondrial function. Nat Commun. 2018 Aug 24;9(1):3399).

Additional NFKB inhibitors

Two additional NF-κB inhibitors have been tested on MCD-induced Jmjd3 expression. Once again, NF-κB inhibitors completely abolished MCD-induced Jmjd3 expression (Supplementary Figure 2).

3) Demonstrate in vivo relevance of the observed statin effects in order to justify that this is indeed a mechanism by which statins can affect inflammation

To understand the function of statins in vivo, we need to go back to some of the basic cell biology and lipid regulation.

About 50% of the total body cholesterol is derived from de novo synthesis in humans. The primary site of endogenous cholesterol biosynthesis is the liver, where statins inhibit cholesterol synthesis as their primary pharmacological action. This blockade of cholesterol production in the liver activates a compensatory mechanism: the upregulation of hepatic LDL-receptors. The liver then increases LDL uptake from the circulation to acquire cholesterol, thereby lowering plasma LDL levels.

In addition, statins can decrease cholesterol content in extrahepatic tissues by two processes.

1) Lowering the circulating LDL by statins would decrease cholesterol content of peripheral tissues/cells *--* LDL has evolved to transport cholesterol (which is insoluble in plasma) to supply cholesterol to the peripheral tissues (Brown MS, Kovanen PT, Goldstein JL. Evolution of the LDL receptor concept-from cultured cells to intact animals. Ann N Y Acad Sci. 1980; 348:48-6) The uptake of cholesterol into peripheral cells is through LDL receptor-mediated endocytosis. LDL receptors are widely expressed on the plasma membrane of nearly all peripheral tissues in animals, including blood immune cells (Kovanen PT, Basu SK, Goldstein JL, Brown MS. Low density lipoprotein receptors in bovine adrenal cortex. II. Low density lipoprotein binding to membranes prepared from fresh tissue. Endocrinology. 1979 Mar;104(3):610-6.)**.** Therefore, cholesterol content of peripheral cells is directly associated with circulating LDL levels.

2) Statins could directly act on the peripheral tissues to suppress cholesterol biosynthesis -- Statins taken orally reach the liver though the circulation, which inevitably results in significant concentrations of statins in the plasma. The pharmacological effects of statins on peripheral cells depend on the half-life and dosage of statins and also on hydrophobicity of statin varieties. Nevertheless, statins are extremely potent (Ki = 0.2 to 1 nM) (Endo A, Hasumi K. HMG-CoA reductase inhibitors. Nat Prod Rep. 1993 Dec;10(6):541-50) and could significantly interfere with cholesterol synthesis in extrahepatic tissues. In addition, bone-marrow derived immune cells are highly proliferative and hence have a high demand for cholesterol, which makes them particularly susceptible to the inhibitory effects of statins. This is confirmed by studies that show that statins can inhibit cholesterol synthesis in circulating macrophages (Chow OA et al., Statins enhance formation of phagocyte extracellular traps. Cell Host Microbe. 2010 Nov 18;8(5):445-54).

We speculate that it is the reduction of cholesterol content in immune cells by statins that underlines the anti-inflammatory functions of statins. Thus, our in vitro studies reported here constitute an important first step and a proof-of-principle: lowering cholesterol content in macrophages leads to an anti-inflammatory state in macrophages.

We agree with the reviewers that the definitive test is an in vivo intervention with human subjects. For example, in matched individuals taking statin or not, macrophage/ cholesterol/Jmjd3 would be evaluated. Inflammatory responses to various stimuli would then follow. However, a clinical trial of this nature is beyond the scope of our current study. We hope that publishing the data in our manuscript will be a spark for readers to consider such in vivo studies as a future goal.

As for in vivo mouse models, the situation here is unique. Mice intrinsically have almost no LDL in the circulation and the major lipoprotein is HDL (unlike humans). In order to study the role of plasma cholesterol in atherosclerosis, applicable to humans, LDLr^-/-^ or ApoE^-/-^ mouse models (apoE is an LDL associated protein and binds to LDLr to enable LDL uptake) were established. These mouse models have elevated levels of plasma LDL, similar to that in human, and have greatly contributed to our knowledge of atherosclerosis. However, statins could not be used in these mouse models to lower plasma cholesterol, since there is no LDLr nor apoE -mediated uptake of LDL at the liver in these animals.

4) Provide details on which statins were used, which concentrations, and treatment durations

Stains names, concentrations and treatment conditions are added in figure legends.

Reviewer #1 (Recommendations for the authors):1. in vivo treatment with statins (either humans or animal models) versus control, followed by isolation of macrophages to assess gene expression, mitochondrial function, and mitochondrial membrane composition would be important to support the core conclusions

See responses to Essential revision 3.

2. Instead of using CCCP, selective shRNA depletion of ATP synthase would provide more targeted information without potential mitochondrial toxicity and ROS generation that occurs with CCCP.

See responses to Essential revision 2.

3. The RNA-seq and ATAC-seq data are presented in a very limited and superficial manner. Instead of showing only the inferred transcription factor binding sites, it would be helpful to first show the most upregulated and downregulated genes and gene expression pathways in the RNA-seq data. Similar in-depth analysis of the ATAC-seq data would be helpful, and further determination of concordance versus discordance between the RNA-seq and ATAC-seq data. Importantly, as H3K27 methylation is a key part of the conclusions, it would be important to perform ChIP-seq for H3K27 methylation instead of selected PCR. This would provide a comprehensive insight into which gene programs are affected. These in-depth bioinformatic analyses should be accompanied by the descriptions in the Methods sections

RNA-seq data was re-analyzed by Gene Set Enrichment Analysis (GSEA) on all differentially expressed genes, both up and down. This analysis is unsupervised, i.e. no expectation whether genes would go up or go down by statins or MCD. This showed the NF-κB pathway is the dominant process activated by statin or MCD (Figure 1).

Similar GSEA on ATAC-seq gave identical conclusion (Figure 6 A), which is consistent with RNA-seq GSEA (Figure 1 B). The examples of IL-1b and TNF-a were added (Figure 6 B).

The current study focused on established anti-inflammatory genes, IL-10, Arg1, Ym1, and Mrc1. Their expressions are enhanced by statin. ChIP-PCR of H3K27me3 supplied a potential mechanism for this enhancement, which is sufficient at this stage of study. On another hand, ChIP-seq for H3K27me3 would provide genomic information of H3K27me3 distribution, including these mentioned above. It would be a big and complex undertaking and would also require RNA-seq of LPS and IL-4 RNA-seq to be meaningful. This is, in our opinion, beyond the scope of current study but in the future.

4. Statin experiments need to indicate which statins were used, which concentrations, and how long cells were treated. This is essential to ensure reproducibility and comparison with other statin studies.

The specific statin, dosage and incubation time used in each experiment have been added to figure legends.

Reviewer #2 (Recommendations for the authors):The authors would benefit from thinking about their experiments and what they do or do not clearly show. It is clear that JMJD3 is being upregulated by statins and MCD however, some of the mechanistic claims that they are making feel a little too preliminary.For example, MG-132 is a well-characterized proteasome inhibitor. There may be a better NFKB inhibitor to use here for this analysis.

Two additional NF-κB inhibitors were studied with identical effects on MCD-induced JMJD3 expression (Supplementary Figure 2).

The western blots in Figure 1 could be of slightly better quality to really convincingly show that these cholesterol perturbations are changing protein expression. Furthermore, for the TNF and IL1b results, they would be strengthened by showing these perturbations have an impact at the protein level.

For logistical reasons, repeating the western blots is not something we can go back to at the moment. I hope the reviewer will agree they are good enough as is.

We are also not in a position to add data at the protein level. At this stage of our investigations, the mechanism we are examining is at the stage of regulation of gene expression. However, I agree with the reviewer this is a limitation and we have added a comment to the manuscript to state this (P8, line 10): “Of note, this current study focuses mostly on the regulation of gene expression. The ultimate impact of statins on inflammation should be extended and confirmed at protein level in future studies.”

The authors also don't really discuss the literature about cholesterol and the inflammasome which may be contributing to the results that they are seeing with IL1b.

In terms of cholesterol levels and the inflammasome, it appears that the inflammasome is activated by excess cholesterol, not by cholesterol reduction (PMID: 25614320). Also, inflammasome activation does not necessarily leads to IL1b expression, as recently reviewed by Tall et al. (PMID: 37228237). In addition, the magnitude of change in IL1b by MCD or statin is extremely low, compared with LPS (Figure 2B, b), suggesting a mechanism that is unrelated to inflammation.

There are several figures that don't convey a clear message. For example, the presentation of the omics results could be improved in order to better convey a strong message. While it is clear that the statins impact the ATACseq result, whether this is at all related to altered JMJD3 expression is not clear so figure 6 in its current form is divorced from the narrative that was established in the earlier figures.

Sub-titles are added to more clearly convey the conclusions for each figure.

We agree that changes in ATAC-seq are not entirely due to JMJD3. We showed that NF-κB is upstream of JMJD3 upregulation. NF-κB could alter epigenome, as has been well documented by other studies. Jmjd3, by removal of H3K27me3, should further alter ATAC -seq. Therefore, the change in ATAC-seq by cholesterol reduction is consistent with NF-κB activation and JMJD3 upregulation. This change prompted us to hypothesize that cholesterol reduction changes epigenome and poises genes for alternative expression when activated. We therefore investigated LPS and IL-4 stimulated gene expression.

All small molecule perturbation experiments would benefit from (1) controls to show that the drugs are not toxic at the concentration used and (2) evidence that these results are specific for the target of interest and not just wholesale altering gene expression across the transcriptome.

Toxicity tests have been added.

[Editors' note: further revisions were suggested prior to acceptance, as described below.]

The manuscript has been improved but there are some remaining issues that need to be addressed, as outlined below:Essential revisions that were requested during the last review have not been addressed. This includes some in vivo data to support the key claims of the work as well as a standard bioinformatic analysis which shows all the differentially expressed genes (top 25 or 50 can be shown in heatmaps, full table can be a supplement). The answers that were provided to the essential revisions do not adequately address the reviewer comments. The interpretations and conclusions of the manuscript do not match the data that are presented.Please review the comments and essential revisions requested from the 1st review and also take into account the comments provided again below:Reviewer 1 (Recommendations for the authors):1. There are no in vivo studies in the revision even though demonstrating in vivo relevance of the findings would be essential to bolster the claims of the work. There are several animal models available in which statin efficacy has been demonstrated

We would like to thank this reviewer for insisting on the need for in vivo studies, which truly do elevate the impact of the study.

We have performed statin-feeding in wt mice on chow diet and analyzed freshly isolated peritoneal macrophages. As shown in Figure 9, 14 days of statin-feeding (simvastatin, 100 mg/kg/day, 14 days) led to about 20% decrease in peritoneal macrophage cholesterol and upregulation of the expression of *Jmjd3*. Inflammatory responses to LPS and IL4 were also identical to those of BMDMs when they were treated with statins in vitro.

We believe these new in vivo data bolster the conclusions of our manuscript.

2. The bioinformatic analysis does not show the various differentially regulated genes nor is there an in-depth analysis of the epigenetic data that was recommended as essential. In-depth analyses recommended listing all the differentially expressed genes – not just selected pathways via GSEA – for RNA, differentially accessible chromatin regions, etc. GSEA just gives a limited perspective and does not convey a sense of the overall data.

Following data have been added to supplementary files:

Excel file of differentially regulated genes (RNA-seq) by statin: Supplementary file 1.

Excel file of differentially regulated genes (RNA-seq) by MCD: Supplementary file 2.

Excel file of genes with altered accessibilities to transposase (ATAC-seq) by MCD: Supplementary file 3.

List of genes shared in ATAC-seq (Supplementary file 2) and RNA-seq (Supplementary file 3): Supplementary file 4.

Excel file of differentially regulated genes (RNA-seq) in LPS and statin/LPS stimulated cells: Supplementary file 5.

In addition, the following has been added to supplementary figures:

Top 40 upregulated genes by statin: Figure 1—figure supplement 1, A, a.

Top 40 upregulated genes by MCD: Figure 1—figure supplement 1, A, b.

Genes down-regulated by statin or MCD are analyzed for TF enrichment: Figure 1—figure supplement 2, C.

Reviewer #2 (Recommendations for the authors):The authors have addressed many of my concerns; however, I suggest they review their language closely in their manuscript one more time to make sure that they are pursuing conservative interpretations of their data without overclaiming.

Text is edited to be more conservative in interpretations of the data and restrain on speculations. For example:

“We speculate that low level of activation of NF-κB could be part of homeostatic regulations.” (P4) is removed.

“However, statin also lowered the maximal respiration (Figure 3B, *c*), possibly due to the 2-day treatment period required for the statin treatment. Overall, statin or MCD suppresses ATP synthase in macrophages.”(P5)

Reviewer #3 (Recommendations for the authors):Although the findings are potentially of interest to eLife readership, regrettably the revised manuscript and author responses fail to address the comments pointed out by this reviewer.

As none of the comments from reviewer 3 appeared in essential revision requests, we did not specifically address the comment about RNA-seq and ATAC-seq. Now we have added several files/figures to address the second comment of reviewer 1 above. We believe that these also sufficiently address the comments from reviewer 3.